# Cohort study of high maternal body mass index and the risk of adverse pregnancy and delivery outcomes in Scotland

Lawrence Doi ,[1] Andrew James Williams ,[2] Louise Marryat ,[3,4] John Frank[3,5]

For numbered affiliations see end of article.

**Correspondence to**
Dr Lawrence Doi;
larry.doi@ed.ac.uk

## ABSTRACT

**Objective** To examine the association between high maternal weight status and complications during pregnancy and delivery.

**Setting** Scotland.

**Participants** Data from 132 899 first-time singleton deliveries in Scotland between 2008 and 2015 were used. Women with overweight and obesity were compared with women with normal weight. Associations between maternal body mass index and complications during pregnancy and delivery were evaluated.

**Outcome measures** Gestational diabetes, gestational hypertension, pre-eclampsia, placenta praevia, placental abruption, induction of labour, elective and emergency caesarean sections, pre-term delivery, post-term delivery, low Apgar score, small for gestational age and large for gestational age.

**Results** In the multivariable models controlling for potential confounders, we found that, compared with women with normal weight, the odds of the following outcomes were significantly increased for women with overweight and obesity (overweight adjusted ORs; 95% CI, followed by the same for women with obesity): gestational hypertension (1.61; 1.49 to 1.74), (2.48; 2.30 to 2.68); gestational diabetes (2.14; 1.86 to 2.46), (8.25; 7.33 to 9.30); pre-eclampsia (1.46; 1.32 to 1.63) (2.07; 1.87 to 2.29); labour induction (1.28; 1.23 to 1.33), (1.69; 1.62 to 1.76) and emergency caesarean section (1.82; 1.74 to 1.91), (3.14; 3.00 to 3.29).

**Conclusions** Women with overweight and obesity in Scotland are at greater odds of adverse pregnancy and delivery outcomes. The odds of these conditions increases with increasing body mass index. Health professionals should be empowered and trained to deliver promising dietary and lifestyle interventions to women at risk of overweight and obesity prior to conception, and control excessive weight gain in pregnancy.

## Strengths and limitations of this study

► This study used a large, retrospectively accessed but cohort-structured, national database covering some of the major maternal and neonatal outcomes in Scotland over eight recent years.

► Analyses were adjusted for some of the key potential confounders to estimate impact of high maternal-weight status on each outcome.

► All women with body mass index (BMI) of 30 kg/m$^2$ or more were considered as having obesity; it is likely that differentiating morbid obesity or obesity class II and III from obesity would have generated more precise estimates.

► The completeness of the recording of BMI increased during the study period (2008 to 2015) from 69% to 98%. Using data from the earlier years when the BMI was missing more often might have biassed the study sample if it was the case that BMI was not missing at random.

## INTRODUCTION

The increasing global prevalence of overweight and obesity makes it more likely that a growing number of women with high body mass index (BMI) are becoming pregnant. High maternal BMI during pregnancy has immediate implications for pregnancy complications as well as long-term health implications for both women and offspring.[1,2] For instance, in terms of pregnancy complications, a systematic review and meta-analysis involving 11 cohort studies found that caesarean delivery risk increased by 50% in pregnant women who were overweight and was more than double for women who were obese compared with women with normal BMI.[3] High BMI during pregnancy could lead to future chronic disease such as diabetes, heart disease and hypertension.[4] Surviving offspring are also more prone to long-term obesity, hypertension, coronary heart disease, diabetes, stroke and asthma.[4,5]

Both immediate and long-term health implications of high BMI during pregnancy have economic consequences. For example, a recent study examining infant health utilisation and costs on the National Health Service (NHS) in the UK of infants born to women with overweight or obesity found that total mean additional resource cost for infants born to women

who are overweight was £65.13, and £1138.11 for infants born to women who are obese.[6]

Maternal weights are currently high in Scotland: a recent study reported that 31.5% of mothers were overweight and a further 23.6% were affected by obesity.[7] A retrospective cohort study using Scottish obstetrical data from 2003 to 2010 examining the impact of maternal BMI on clinical complications, inpatient admissions and additional short-term costs to the NHS in Scotland revealed that maternal BMI influences maternal and neonatal morbidity, the number and duration of maternal and neonatal admissions and health service costs.[8] The study also showed that in comparison with women of normal weight, women who were overweight, obese or severely obese had an increased risk of essential hypertension (1.87 (1.18 to 2.96), 11.90 (7.18 to 19.72) and 36.10 (18.33 to 71.10)), pregnancy-induced hypertension (1.76 (1.60 to 1.95), 2.98 (2.65 to 3.36) and 4.48 (3.57 to 5.63)), gestational diabetes (3.39 (2.30 to 4.99), 11.90 (7.54 to 18.79) and 67.40 (37.84 to 120.03)), emergency caesarean section (1.94 (1.71 to 2.21), 3.40 (2.91 to 3.96) and 14.34 (9.38 to 21.94)) and elective caesarean section (2.06 (1.84 to 2.30), 4.61 (4.06 to 5.24) and 17.92 (13.20 to 24.34)).[8] Smith *et al*,[9] using data from a retrospective cohort study of 187 290 women in Scotland to examine the risk of maternal obesity in early pregnancy and the risk of pre-term delivery, found that among nulliparous women, the risk of an elective pre-term delivery increased with increasing BMI. The study also observed that 40% of morbidly obese nulliparous women who experienced an elective pre-term delivery had been diagnosed with pre-eclampsia, in contrast with only 2.6% of the remaining study population.[9] Maternal obesity has also been linked to low Apgar score and pre-term and post-term delivery as well as the risk of intrapartum complications, such as placenta praevia and placental abruption.[2 10] It is likely that any risk of intrapartum complications may necessitate labour induction or more frequent caesarean delivery.

In the current study, we hypothesised, based on previous studies elsewhere, that women with obesity and their babies experience higher rates of virtually all perinatal complications, which are routinely collected in Scotland, except perhaps low birth weight (due to the macrosomia effect of overt or covert gestational diabetes), and that women with overweight and their babies experience an excess risk of these same outcomes, but one not as high as women with obesity and their offspring. Therefore, our aim was to use more recent data to examine the associations between high maternal BMI and complications during pregnancy and delivery in Scotland. Understanding of these associations can highlight areas where prevention strategies could be targeted.

## METHODS

### Study population and data sources

This retrospective cohort study used data from 132 899 first-time mothers who gave birth to only one child in Scotland between January 2008 and December 2015. The women and infants were identified within three electronic medical record databases: the Scottish Morbidity Record (SMR) 01 and 02 and Scottish Birth Record (SBR). SMR01 is generated for patients receiving inpatient or day care in the General or Acute specialties, while SMR02 is generated for patients receiving inpatient or day care in the Obstetrical Specialities. The SBR records all of a baby's neonatal care in Scotland. Relevant outcome variables are recorded in these databases according to the WHO's International Classification of Diseases, Tenth Revision (ICD-10) or NHS Scotland classifications.[11] Further description of the content of these databases is available[12] and in the online supplementary file 1. The study was designed as a clinical audit so did not require approval from a Research Ethics Committee. However, approval was obtained from the Public Benefit and Privacy Panel via the national Electronic Data Research and Innovation Service to use the anonymised data collected by these registries. As a clinical audit making secondary use of anonymised electronic patient records, it was necessary to account for missing data which was relevant to this research study (see the online supplementary file 2 for the flow diagram illustrating how the final sample size was reached). The large number of variables involved in this study and a low likelihood that missingness was at random meant that imputation methods would have been complicated and a complete case analysis was more suitable for this population-wide study.

### Patient and public involvement

Patients were not involved in the design, analyses and interpretation of this study.

### Exposure variable

More than 80% of pregnant women in Scotland present themselves for antenatal care during the first trimester of their pregnancy.[13] Height and weight are usually measured by the midwife at the first antenatal visit, typically before 12 weeks of pregnancy. BMI was calculated using the formula weight (kg)/height (m$^2$). BMI categories were defined as normal (<25 kg/m$^2$), overweight (≥25 kg/m$^2$ to <30 kg/m$^2$) and obese (≥30 kg/m$^2$). BMI completeness was 69% in 2008 but this increased gradually to 87% in 2011 when recording of weight and height became mandatory. By 2015 BMI completeness was 98%.

### Outcomes

Outcome measures included were maternal or pregnancy complications organised into three groups related to when they occur during the pregnancy;

► Conditions affecting pregnancy: gestational diabetes, gestational hypertension and pre-eclampsia (high blood pressure and protein in urine).

► Conditions affecting delivery: placenta praevia (when a baby's placenta partially or totally covers the mother's cervix), placental abruption (when the placenta separates early from the uterus before childbirth),

**Table 1** Maternal characteristics among normal weight, overweight and obese women* (singleton, first pregnancies)

| | Normal† n=71 538 | | Overweight† n=36 188 | | Obese† n=25 173 | |
|---|---|---|---|---|---|---|
| | N | % | N | % | N | % |
| **Maternal age (y)** | | | | | | |
| 20–24 | 19 372 | 27.1 | 9152 | 25.3 | 6851 | 27.2 |
| 25–29 | 23 871 | 33.4 | 11 895 | 32.9 | 8280 | 32.9 |
| 30–34 | 20 488 | 28.6 | 10 304 | 28.5 | 6777 | 26.9 |
| 35–39 | 7807 | 10.9 | 4837 | 13.4 | 3265 | 13.0 |
| **Carstairs 2001 quintiles for Scotland** | | | | | | |
| Q1 (least deprived) | 14 546 | 20.3 | 6715 | 18.6 | 3833 | 15.2 |
| Q2 | 13 574 | 19.0 | 6733 | 18.6 | 4578 | 18.2 |
| Q3 | 14 245 | 19.9 | 7382 | 20.4 | 5005 | 19.9 |
| Q4 | 14 930 | 20.9 | 7777 | 21.5 | 5909 | 23.5 |
| Q5 (most deprived) | 14 243 | 19.9 | 7581 | 21.0 | 5848 | 23.2 |
| **Maternal smoking in pregnancy** | | | | | | |
| No | 61 116 | 85.4 | 31 119 | 86.0 | 21 130 | 83.9 |
| Yes | 10 422 | 14.6 | 5069 | 14.0 | 4043 | 16.1 |

*Figures show women who had complete data on age, deprivation, maternal smoking and the conditions occurring during pregnancy.
†Maternal weight status at first antenatal visit.

pre-term delivery (defined as less than 37 weeks of gestation), post-term delivery (more than 42 weeks of gestation), small for gestational age (SGA) and large for gestational age (LGA). SGA were infants with birth weight of ≤10th percentile for gestational age according to UK1990 growth reference curve,[14 15] and those with LGA were infants with birth weight ≥90th percentile.

► Delivery: induction of labour, caesarean delivery (includes elective and emergency caesarean sections) and low Apgar score (less than '7' at 5 min).

## Covariates

Maternal age at delivery, smoking during pregnancy and Carstairs 2001 quintiles for socioeconomic status in Scotland, based on the postcode of the mother's residence at birth, were considered as potentially confounding variables and were included as covariates in the adjusted analyses. Table 1 describes the covariates used in this study by maternal weight status among singleton (a pregnancy with one foetus, as opposed to twins or multiples) first-time pregnancies. The data in table 1 show the numbers of women who had data on age, deprivation, maternal smoking and the three conditions being studied that occur during pregnancy.

## Data analyses

Using Stata 14,[16] logistic regression models were fitted to calculate ORs. BMI groups with overweight and obesity were compared with the normal BMI group (the reference population). A CI of 95% was produced for all ORs. The analyses of the outcomes proceeded in a systematic approach.

The outcomes were analysed in the three groups described above. As some of the outcomes were mutually exclusive (eg, a baby cannot be both small and large for gestational age) those with the opposing outcome were excluded from the outcome being analysed. Each model was also adjusted for any of the outcomes that occurred earlier in the pregnancy. Table 2 provides information on the covariates adjusted for in each model.

## RESULTS

Within our study population 53.8% of pregnant women were categorised as normal weight, 27.2% as overweight and 18.9% as obese. The socio-demographic characteristics of the women in the three BMI categories are presented in table 1. Maternal smoking prevalence was slightly higher among women with obesity than in women with normal weight or overweight. Among the women who were overweight, 21.0% were from the most deprived group and 18.6% were from the least deprived group. However, the difference in social deprivation was more marked within women with obesity. Among this group, 23.2% were from the most deprived group while 15.2% were from the least deprived group.

Table 3 shows ORs for pregnancy and delivery complications, among women who were overweight or obese. The odds of gestational diabetes, pre-eclampsia and hypertension increased steadily with increasing BMI. Compared with the normal BMI group, the OR of gestational diabetes was 2.14 (95% CI: 1.86 to 2.46) but among women who were obese the OR increased to 8.25 (95% CI: 7.33 to 9.30). Relative to women who were of normal

**Table 2** Full list of variables adjusted for in each of the models in table 3

| | Risk factors | | | |
| --- | --- | --- | --- | --- |
| | **Maternal circumstances** | **Conditions affecting pregnancy** | **Conditions affecting delivery** | **Delivery** |
| **Conditions affecting pregnancy** | | | | |
| Gestational hypertension | ► Age<br>► Deprivation<br>► Smoking status<br>► Weight status | ► Gestational diabetes<br>► Pre-eclampsia | – | – |
| Gestational diabetes | ► Age<br>► Deprivation<br>► Smoking status<br>► Weight status | ► Gestational hypertension<br>► Pre-eclampsia | – | – |
| Pre-eclampsia | ► Age<br>► Deprivation<br>► Smoking status<br>► Weight status | ► Gestational hypertension<br>► Gestational diabetes | – | – |
| **Conditions affecting delivery** | | | | |
| Placenta praevia | ► Age<br>► Deprivation<br>► Smoking status<br>► Weight status | ► Gestational hypertension<br>► Gestational diabetes<br>► Pre-eclampsia | ► Placental abruption<br>► Size for gestational age<br>► Full, pre-term or post-term | – |
| Placental abruption | ► Age<br>► Deprivation<br>► Smoking status<br>► Weight status | ► Gestational hypertension<br>► Gestational diabetes<br>► Pre-eclampsia | ► Placenta praevia<br>► Size for gestational age<br>► Full, pre-term or post-term | – |
| Small for gestational age | ► Age<br>► Deprivation<br>► Smoking status<br>► Weight status | ► Gestational hypertension<br>► Gestational diabetes<br>► Pre-eclampsia | ► Placental abruption<br>► Placenta praevia<br>► Full, pre-term or post-term | – |
| Large for gestational age | ► Age<br>► Deprivation<br>► Smoking status<br>► Weight status | ► Gestational hypertension<br>► Gestational diabetes<br>► Pre-eclampsia | ► Placental abruption<br>► Placenta praevia<br>► Full, pre-term or post-term | – |
| Pre-term | ► Age<br>► Deprivation<br>► Smoking status<br>► Weight status | ► Gestational hypertension<br>► Gestational diabetes<br>► Pre-eclampsia | ► Placental abruption<br>► Placenta praevia<br>► Size for gestational age | – |
| Post-term | ► Age<br>► Deprivation<br>► Smoking status<br>► Weight status | ► Gestational hypertension<br>► Gestational diabetes<br>► Pre-eclampsia | ► Placental abruption<br>► Placenta praevia<br>► Size for gestational age | – |
| **Delivery** | | | | |
| Induction of labour | ► Age<br>► Deprivation<br>► Smoking status<br>► Weight status | ► Gestational hypertension<br>► Gestational diabetes<br>► Pre-eclampsia | ► Placental abruption<br>► Placenta praevia<br>► Size for gestational age<br>► Full, pre-term or post-term | – |
| Caesarean section | ► Age<br>► Deprivation<br>► Smoking status<br>► Weight status | ► Gestational hypertension<br>► Gestational diabetes<br>► Pre-eclampsia | ► Placental abruption<br>► Placenta praevia<br>► Size for gestational age<br>► Full, pre-term or post-term | – |
| Emergency caesarean section | ► Age<br>► Deprivation<br>► Smoking status<br>► Weight status | ► Gestational hypertension<br>► Gestational diabetes<br>► Pre-eclampsia | ► Placental abruption<br>► Placenta praevia<br>► Size for gestational age<br>► Full, pre-term or post-term | – |

Continued

**Table 2** Continued

| | Risk factors | | | |
| | **Maternal circumstances** | **Conditions affecting pregnancy** | **Conditions affecting delivery** | **Delivery** |
|---|---|---|---|---|
| Apgar score | ► Age<br>► Deprivation<br>► Smoking status<br>► Weight status | ► Gestational hypertension<br>► Gestational diabetes<br>► Pre-eclampsia | ► Placental abruption<br>► Placenta praevia<br>► Size for gestational age<br>► Full, pre-term or post-term | ► Mode of delivery |

weight, the adjusted OR of pre-eclampsia for women who were overweight was 1.46 (95% CI: 1.32 to 1.62), and 2.07 (95% CI: 1.87 to 2.29) for women who were obese. The OR of gestational hypertension, compared with women with normal weight, was 1.61 (95% CI: 1.49 to 1.74) for women with overweight, and 2.48 (95% CI: 2.30 to 2.68) for women with obesity.

Regarding conditions affecting delivery, the OR of placenta praevia was not statistically significantly different for both women who were overweight (OR 1.23, 95% CI: 0.90 to 1.68) or obese (OR 0.81, 95% CI: 0.54 to 1.22), compared with women with normal weight. The OR of experiencing placental abruption was also not statistically significantly different across the different BMI categories.

In contrast with the normal BMI group, births to women who were overweight and obese were associated with decreased odds of small-for-gestational age ORs 0.81 (95% CI: 0.78 to 0.85) and 0.79 (95% CI: 0.74 to 0.83), respectively. However, the odds of large-for-gestational age newborns increased among women with overweight, OR of 1.27 (95% CI: 1.23 to 1.30) and women with obesity, OR of 1.53 (95% CI: 1.48 to 1.58), compared with women of normal weight. Regarding the ORs for the pre-term and post-term outcomes, only the pre-term outcome for the obese group was statistically significant and the others were not significant: compared with the normal BMI group, the adjusted OR of pre-term delivery was 1.02 (95% CI: 0.96 to 1.07), however among women who were obese the OR was 1.11 (95% CI: 1.05 to 1.18). Relative to women who were of normal weight, the adjusted OR of post-term for women who were overweight was 1.57 (95% CI: 0.93 to 2.68) and 1.47 (95% CI: 0.78 to 2.77) for women who were obese.

The odds of induction of labour and caesarean section, either elective or emergency, increased with increasing BMI. Regarding induction of labour, the ORs were statistically significant for women with overweight (OR 1.28, 95% CI: 1.23 to 1.33) and those with obesity (OR 1.69, 95% CI: 1.62 to 1.76) compared with women with normal weight. Women who were overweight had ORs of 1.34 (95% CI: 1.29 to 1.39) for having an elective Caesarean section and higher ORs (1.82, 95% CI: 1.74 to 1.91) for undergoing emergency Caesarean section, compared with women of normal weight. The corresponding ORs for women with obesity were 1.80 (95% CI: 1.73 to 1.88)

and 3.14 (95% CI: 3.00 to 3.29). Being overweight or obese was associated with reduced odds of low Apgar score. This was barely statistically significant for women who were overweight (OR 0.95, 95% CI: 0.92 to 0.99) or obese (OR 0.96, 95% CI: 0.93 to 1.00).

## DISCUSSION

In this large, retrospective cohort study, we found that overweight or obesity during pregnancy was associated with increased odds of several adverse pregnancy and delivery complications. Aside from obesity, we also examined overweight because in most populations, a greater number of women are overweight rather than obese, so it is important to also understand the impact of overweight on pregnancy and neonatal outcomes.

In terms of the associations between high maternal BMI and conditions that occur *during* pregnancy, we found that the odds of all the conditions considered (gestational hypertension, gestational diabetes and pre-eclampsia) increased steadily with increasing BMI, which is in line with similar studies.[2 10 17] A study compared women of normal weight to women who were morbidly obese (BMI greater than 40), and also found that there was an increased odds of pre-eclampsia (OR 4.82; 95% CI: 4.04 to 5.74).[10] Our study also found that, aside from heightened pre-eclampsia odds for women with obesity, being overweight was also significantly associated with this outcome, although to a lesser degree. A meta-analysis of the association between maternal BMI and the risk of pre-eclampsia showed that the risk doubled with each 5–7 kg/m$^2$ increase in pre-pregnancy BMI.[18] It is evident that the risk of pre-eclampsia increases with the degree of weight gain; therefore preventative strategies should be focussed on getting women, especially those already overweight, to reduce weight prior to conception. Weight loss in pregnancy requires careful management in order to avoid unintended consequences.[19] Nevertheless, women often engage with health professionals during pregnancy; therefore dietary and lifestyle interventions such as physical activity, which have been shown by reviews and meta-analyses[19 20] to reduce gestational weight gain and improve outcomes for both mother and baby, could be provided to them.

Generally, rates of Caesarean delivery have increased significantly across many developed countries in recent

**Table 3** Pregnancy and delivery complications among normal, overweight and obese singleton women

| | Total | Normal | | | Overweight | | | Obese | | |
|---|---|---|---|---|---|---|---|---|---|---|
| | Sample | N (%) | Cases (%) | Adjusted OR* (95% CI) | N (%) | Cases (%) | Adjusted OR* (95% CI) | N (%) | Cases (%) | Adjusted OR* (95% CI) |
| **Conditions affecting pregnancy** | | | | | | | | | | |
| Gestational hypertension | 132 899 | 71 538 (53.8) | 1550 (2.2) | 1.61 (1.49 to 1.74) | 36 188 (27.2) | 1239 (3.4) | 1.61 (1.49 to 1.74) | 25 173 (18.9) | 1275 (5.1) | 2.48 (2.30 to 2.68) |
| Gestational diabetes | 132 899 | 71 538 (53.8) | 377 (0.5) | 2.14 (1.86 to 2.46) | 36 188 (27.2) | 418 (1.2) | 2.14 (1.86 to 2.46) | 25 173 (18.9) | 1082 (4.3) | 8.25 (7.33 to 9.30) |
| Pre-eclampsia | 132 899 | 71 538 (53.8) | 906 (1.3) | 1.46 (1.32 to 1.62) | 36 188 (27.2) | 664 (1.8) | 1.46 (1.32 to 1.62) | 25 173 (18.9) | 640 (2.5) | 2.07 (1.87 to 2.29) |
| **Conditions affecting delivery** | | | | | | | | | | |
| Placenta praevia† | 132 212 | 71 172 (53.8) | 102 (0.1) | 1.23 (0.90 to 1.68) | 36 001 (27.2) | 66 (0.2) | 1.23 (0.90 to 1.68) | 25 039 (18.9) | 30 (0.1) | 0.81 (0.54 to 1.22) |
| Placental abruption† | 132 212 | 71 172 (53.8) | 133 (0.2) | 0.81 (0.59 to 1.11) | 36 001 (27.2) | 55 (0.2) | 0.81 (0.59 to 1.11) | 25 039 (18.9) | 38 (0.2) | 0.76 (0.53 to 1.10) |
| Small for gestational age†‡ | 94 906 | 53 320 (56.2) | 6664 (12.5) | 0.81 (0.78 to 0.85) | 25 122 (26.5) | 2645 (10.5) | 0.81 (0.78 to 0.85) | 16 464 (17.4) | 1726 (10.5) | 0.79 (0.74 to 0.83) |
| Large for gestational age†‡ | 121 177 | 64 508 (53.2) | 17 852 (27.7) | 1.27 (1.23 to 1.30) | 33 356 (27.5) | 10 879 (32.6) | 1.27 (1.23 to 1.30) | 23 313 (19.2) | 8575 (36.8) | 1.53 (1.48 to 1.58) |
| Pre-term†‡ | 132 212 | 71 172 (53.8) | 4295 (6.0) | 1.02 (0.96 to 1.07) | 36 001 (27.2) | 2231 (6.2) | 1.02 (0.96 to 1.07) | 25 039 (18.9) | 1725 (6.9) | 1.11 (1.05 to 1.18) |
| Post-term†‡§ | 78 074 | 43 486 (55.7) | 31 (0.1) | 1.57 (0.93 to 2.68) | 20 973 (26.9) | 24 (0.1) | 1.57 (0.93 to 2.68) | 13 615 (17.4) | 14 (0.1) | 1.47 (0.78 to 2.77) |
| **Delivery** | | | | | | | | | | |
| Induction of labour‡ | 92 967 | 53 617 (57.7) | 13 417 (25.0) | 1.28 (1.23 to 1.33) | 24 342 (26.2) | 7420 (30.5) | 1.28 (1.23 to 1.33) | 15 008 (16.1) | 5712 (38.1) | 1.69 (1.62 to 1.76) |
| Caesarean section‡ | 90 183 | 51 798 (57.4) | 11 598 (22.4) | 1.34 (1.29 to 1.39) | 23 827 (26.4) | 6905 (29.0) | 1.34 (1.29 to 1.39) | 14 558 (16.1) | 5.262 (36.2) | 1.80 (1.73 to 1.88) |
| Emergency caesarean section‡ | 80 938 | 45 715 (56.5) | 5515 (12.1) | 1.82 (1.74 to 1.91) | 21 397 (26.4) | 4475 (20.9) | 1.82 (1.74 to 1.91) | 13 826 (17.1) | 4530 (32.8) | 3.14 (3.00 to 3.29) |
| Apgar score | 129 773 | 70 012 (54.0) | 12 583 (18.0) | 0.95 (0.92 to 0.99) | 35 307 (27.2) | 6125 (17.4) | 0.95 (0.92 to 0.99) | 24 454 (18.8) | 4342 (17.8) | 0.96 (0.93 to 1.00) |

*Adjusted for maternal age, deprivation, smoking in pregnancy and the pre-existing or co-existing conditions (see online supplementary file 2).
†70 post-term births were excluded from these models as none of them experienced the outcome being estimated.
‡These total sample sizes differ as the outcome being estimated is mutually exclusive from one or more of the other outcomes within that group. For example, any baby being delivered pre-term cannot also have been delivered post-term and therefore these two models include the same 'controls' those delivered at term but difference 'cases' pre-term or post-term.
§No participants in the study delivering post-term had placenta praevia, placental abruption or had small or large for gestational age babies; therefore 45 957 participants were dropped from this analyses.

years.[21] Our study found that women with overweight and obesity showed increased odds of Caesarean delivery (both elective and emergency) compared with women with normal weight, but we note that the overall frequency of Caesarean delivery across our obstetric population seems quite high, compared with a previous Swedish study.[10] We also found that women with overweight and obesity are at increased odds of labour induction. A very recent systematic review found that women with obesity are more likely than women with a normal weight to end labour induction with Caesarean delivery.[22] Possible reasons could be that Caesarean delivery is probably less risky now, due to advances in medical science, which facilitate accurate monitoring of the progress of labour and the detection of foetal intrapartum conditions.[23] It is also possible that health professionals in Scotland are intervening earlier with regards to problems in labour among women with overweight or obesity, in order to reduce foetal distress, and its worst outcomes. Nevertheless, this pattern of very high Caesarean rates is concerning for Scotland, which has invested in programmes aimed at promoting natural birth, such as Keeping Childbirth Natural and Dynamic (KCND). The KCND is a maternity care programme introduced by the Scottish Government with the aim of maximising opportunities for women to have as natural a birth experience as possible, reduce unnecessary interventions in low-risk pregnancy and childbirth and to provide women-centred care.[24–26] The early intervention in pregnancy may also explain the reduced odds of low Apgar score for infants born to women with overweight and obesity. It is likely that these women may receive increased monitoring, which means issues can be identified and managed earlier, to reduce any foetal distress in labour.

Adiposity has also been found to increase odds of large-for-gestational age and macrosomia.[27] In this study, we found that births to women with overweight and obesity were associated with increased odds of large-for-gestational age infants, compared with women of normal weight. Excess weight in pregnancy may shift the entire birth weight distribution upwards, perhaps through hormonal mechanisms that operate at lower levels, rather than in full-blown cases of macrosomia in infants of diabetic mothers. It is therefore unsurprising that high maternal BMI significantly decreased the odds of small-for-gestational age among our study population.

We found that pregnant women with obesity were at significantly increased odds of pre-term delivery, however the OR was high but not statistically significant for women with overweight. A systematic review examining the effect of maternal overweight and obesity on pre-term delivery showed that both women with overweight and obesity were at significantly higher risk of pre-term delivery.[28] It has been shown that pre-eclampsia leads to pre-term delivery, especially in elective pre-term delivery.[29] It is not clear why the OR of pre-term delivery for women with overweight in our study population was not statistically significant. However, it is likely that the higher odds of pre-eclampsia in women with obesity, compared with women with overweight, could explain this finding. Regarding post-term delivery, there were no statistically significant ORs among women with both overweight and obesity. As discussed previously, it is likely that early intervention in pregnancy among our study population reduced the odds of the occurrence of post-term delivery.

We examined the association between high maternal BMI and placental abruption and placenta praevia, but found no statistically significant association between each of these two outcomes and overweight or obesity. This finding is congruent with a previous study.[10] It appears that the relationships between maternal overweight and obesity, and both placental abruption and placenta praevia, may require further attention in future research.

### Strengths and limitations

This study comprised a large, retrospectively accessed but cohort-structured, national database, covering several maternal and neonatal outcomes. The analyses used population-wide data with adjustment for some confounders to estimate impact of high maternal-weight status on each outcome. We restricted the analyses to only single births and first pregnancies to ensure that the births in the sample were relatively independent. The data set we were provided with combined underweight and normal weight women as normal BMI group. Using this as the reference group might have strengthened the association between high maternal BMI and the pregnancy and neonatal outcomes considered. However, only a very small number of women are underweight during pregnancy in Scotland in recent years (2.8% in 2018/19).[30] Also, some studies differentiate between different obesity categories, but in this study the data set we accessed did not differentiate these categories, and it was not possible to do this retrospectively; therefore all women with BMI of 30 or more were considered as having obesity. It is likely that differentiating morbid obesity, or obesity class II and III from women with obesity, would have generated additional insight, in the form of a full 'dose response relationship'. The completeness of the recording of BMI increased during the study period (2008 to 2015) from 69% to 98%. Using data from the earlier years, when the BMI was missing more often, might have biassed the study sample if BMI was not missing at random. In addition, mothers aged below 20 years and over 40 years were excluded from analyses due to the low numbers of cases in these age groups with obesity and experiencing adverse outcomes. The study controlled for a limited set of confounders, due to data availability; inclusion of other relevant confounders could have strengthened the analyses. For example, variables such as ethnicity, previous caesarean sections and time of birth were not available in the data set, which we accessed. Also, we could not analyse neonatal outcomes such as stillbirth, neonatal death and congenital anomaly because these outcomes are not completely ascertained in the data set we used.

## CONCLUSION

This study has shown that women who are overweight, and especially those who are obese in Scotland are at greater odds of several pregnancy and delivery complications including gestational hypertension, gestational diabetes, pre-eclampsia, labour induction and Caesarean delivery. The ORs of these conditions increased with increasing BMI. Health professionals should be empowered and trained to deliver promising dietary and lifestyle interventions to women at risk of overweight and obesity prior to conception, and control excessive weight gain in pregnancy.

**Author affiliations**
[1]Scottish Collaboration for Public Health Research and Policy, School of Health in Social Science, University of Edinburgh, Doorway 6, Old Medical School, Teviot Place, Edinburgh, EH8 9AG, UK
[2]European Centre for Environmental and Human Health, College of Medicine and Health, University of Exeter, Knowledge Spa, Royal Cornwall Hospital, Truro, Cornwall, TR1 3HD, UK
[3]Farr Institute at Scotland, Nine, Edinburgh BioQuarter, 9 Little France Road, University of Edinburgh, Edinburgh, EH16 4UX, UK
[4]Centre for Clinical Brain Sciences, Royal Edinburgh Hospital, Kennedy Tower, Morningside Park, University of Edinburgh, Edinburgh, EH10 5HF, UK
[5]Usher Institute of Population Health Sciences and Informatics, Doorway 1, Old Medical School, Teviot Place, University of Edinburgh, Edinburgh, EH8 9AG, UK

**Acknowledgements** The authors are grateful to the Electronic Data Research and Innovation Service of the Information Services Division, NHS National Services Scotland for providing the data used in this paper.

**Contributors** LD, AJW and JF conceived the original idea for the study and obtained the data. AJW led the statistical analyses with support from LD, LM and JF. LD wrote the first draft of the paper and all authors revised successive drafts and approved the final manuscript.

**Funding** This work was funded by the SCPHRP core grant from the Medical Research Council (Grant Number MR/K023209/1) and the Chief Scientist Office of Scotland. AJW is supported by the European Centre for Environment and Human Health, University of Exeter. LM is supported by the Farr Institute @ Scotland, which is supported by a 10-funder consortium: Arthritis Research UK, the British Heart Foundation, Cancer Research UK, the Economic and Social Research Council, the Engineering and Physical Sciences Research Council, the Medical Research Council, the National Institute of Health Research, the National Institute for Social Care and Health Research (Welsh Assembly Government), the Chief Scientist Office (Scottish Government Health Directorates), (MRC Grant No: MR/K007017/1).

**Disclaimer** The funders played no role in the conceptualisation or realisation of the research and no role in the decision to submit it for publication.

**Competing interests** None declared.

**Patient consent for publication** Not required.

**Provenance and peer review** Not commissioned; externally peer reviewed.

**Data availability statement** Data may be obtained from a third party and are not publicly available. Data used was categorised as confidential data release by the Electronic Data Research and Innovation Service of the Information Services Division, NHS National Services Scotland.

**ORCID iDs**
Lawrence Doi http://orcid.org/0000-0001-6853-5050
Andrew James Williams http://orcid.org/0000-0002-2175-8836
Louise Marryat http://orcid.org/0000-0002-6093-4679

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
