## [Reviewer comments · BMJ Open]

ARTICLE DETAILS

TITLE (PROVISIONAL)	A cohort study of high maternal Body Mass Index and the risk of adverse pregnancy and delivery outcomes in Scotland
AUTHORS	Doi, Lawrence; Williams, Andrew James; Marryat, Louise; Frank, John

VERSION 1 – REVIEW

REVIEWER	Richard Derman Thomas Jefferson University, USA
REVIEW RETURNED	12-Oct-2018

GENERAL COMMENTS	Cannot determine if the increase in diabetes (or other outcomes) are the cause of rather than the effect of the reported conclusions.
---

REVIEWER	Dagfinn Aune, Associate Professor Bjørknes University College
REVIEW RETURNED	30-Oct-2018

GENERAL COMMENTS	The authors have conducted a cohort study among 345363 deliveries in Scotland between 2008 and 2015 and investigated the association between maternal BMI and risk of a number of pregnancy outcomes. The authors found increased risk of gestational hypertension, gestational diabetes, preeclampsia, labour induction, and emergency caesarean section among overweight and obese women. The findings are interesting and of great importance and add to a growing body of evidence of the adverse effects of maternal adiposity on pregnancy outcomes. I do have several comments and suggestions below. Page 4 introduction: makes it more likely that a..... Page 4: another reference could be added to the fifth line Aune D, Saugstad OD, Henriksen T, Tonstad S. Maternal body mass index and the risk of fetal death, stillbirth, and infant death: a systematic review and meta-analysis. JAMA. 2014 Apr 16;311(15):1536-46. Table 3. Please add number of cases in total at the left as well as for each category of BMI for each outcome. That would be more informative than the percentages that are provided. Give some more info of what is the comparison for Apgar score. Do you have data regarding very preterm births (before week 32)? Do you have data on macrosomia, infant death, shoulder dystocia, intrauterine growth restriction, neonatal jaundice, neonatal hypoglycemia, resuscitation, wound infection, urinary tract infection, and/or maternal mortality? Because of the limited available data on these
---

outcomes it would have been good to see more data on these outcomes. Would have been great if it would be possible to add results for these outcomes.

It would have been good if the authors could add analyses for more detailed categories of obesity including grade 1 (30-<35), 2 (35-<40) and 3 (40+) in a sensitivity analysis perhaps in a supplementary table. And perhaps also add results using more refined categories of weight BMI <18.5, 18.5-<20.0, 20.0-<22.5, 22.5-<25.0, 25.0-<27.5, 27.5-<30.0, 30.0-<32.5, 32.5-<35.0, 35.0-<40, 40+. The study is very large (and the number of cases must be huge as well) - especially when looking at the ORs and CIs in table 3. Most previous studies have only used the WHO classification of weight, but using more refined categories can be important in determining the optimal level of BMI in pregnancy. Therefore it would be important to provide results with more detailed categories.

Page 11, 3rd paragraph: Being overweight or obese significantly decreased the risk of low Apgar score.....

Page 12: it says that most previous studies have focused on obese women. I'm not so sure if this is a good point or even true as there are a large number of studies that have analysed both overweight and obesity. Same paragraph: women are likely to be overweight rather than obese,

It could also be argued that many women are on the high end of the normal weight range and it is unknown what is the optimal BMI. Clarifying if there is increased risk even within the high-normal BMI range might move the existing literature even further and contribute to better weight recommendations for pregnant women.

Page 14, Discussion, paragraph 4: The lack of association between overweight and stillbirth is almost certainly because of lack of power (OR 1.24 (0.94-1.64) so this should be mentioned as a possible explanation here. Also, even if some of the previous studies have varied in the results for overweight the meta-analysis below detected a clear association between increasing weight even from high normal BMI, to overweight and obesity. Aune D, Saugstad OD, Henriksen T, Tonstad S. Maternal body mass index and the risk of fetal death, stillbirth, and infant death: a systematic review and meta-analysis. JAMA. 2014 Apr 16;311(15):1536-46.

Page 14, paragraph 5 on overweight/obesity and risk of SGA and preterm birth. The inverse association with SGA has been found in many other studies, and should not be considered unexpected as adiposity increases risk of LGA and macrosomia. For preterm birth the results have been mixed with positive, null and inverse association reported, however, if you look at studies on very preterm birth (<34 or <32 weeks) the results are more consistent with an increased risk. For this reason I asked in one of my previous comments if you could also add results for very preterm births.

One thing I am wondering about is if the authors by using the staged approach (sequentially excluding subjects with the previously listed conditions when analyzing each outcome) are introducing some sort of bias or selection effect to the results or

	that the results may become less and less generalizable the further down the table you get. For example maternal adiposity increases the risk of gestational hypertension/preeclampsia and gestational diabetes which all may contribute to the increased risk of stillbirth observed with maternal adiposity. I wonder if the reported associations therefore might be more modest (although still quite strong) than if the full population was used. Are the results very different if using the full study population without making such exclusions? Ref. 7: Heslehurst
--	--

REVIEWER	Jérémy Boujenah Hôpitaux Universitaires Paris Seine Saint-Denis
REVIEW RETURNED	26-Feb-2019

GENERAL COMMENTS	Comments General comments The authors adress the issue of BMI and several outcomes. The main limit of the study is the design :  1. The amount of outcomes limit the interpretation of the study and the scope of the results 2. Important counfounding factors are not included in the analysis and therefore make the results unconclusive or imprecise : such as :  - Previous diabetes - Chronic Hypertension - Previous bariatric surgery - Weight gain during pregnancy - For multiparous women with previous hypertension or Preeclampsia, use of prophylactic aspirin - Spontaneous or ART pregnancy - Types of Fetal abndormalities 3. Uncompleted data about BMI, the gestational age of the first consultation 4. Sub-class of obsesity are not differentiate 5. As writtent by the authors, women could be included more than one time if they had more than one birth. 6. Mode of screening for gestationnal diabetes should be described 7. For each outcomes, prevalence should be given in the group control 8. For condition occuring during pregnancy, antepartum or post partum diagnosis should be precised. 9. The stage fashion accoriding to the clinical timelines is not accurate : Stage 3 is dependant of stage 5 (for example, induction of labor is more frequent in case of SGA...) 10. Induction of labor should be described (reasons). Emergency caesarean section should be separated between planned cesarean section and cesarean section during trial of labor. 11. Collider and confoundings factors in such analysis could be described with a directed acyclic graph It would be more appropriate to adress the issue of one only one outcomes so that to identify all factors that could contribute to the result. One question for one response in nulliparous only.
--

	Confounding bias, collider stratification, and selection biases, are important when addressing issues in obstetrics and perinatal epidemiology. Therefore when reading this paper, results cannot be used in daily practice. Some specific comments Association between stillbirth and obesity is not proven Reference 2 is not appropriate The decreased risk of SGA and preterm delivery with overweight could be the result of collider factors. Protective effects cannot be explained by this study. This paradox may often reflects a statistical problem This paper could be submitted again as a new submission with:  - restricting analysis to one outcome such as : risk of preeclampsia in overweight nulliparous women (it is an example) - Caution should be given to confounding factors when addressing issue in perinatality - Without missing data - With data on previous maternal condition as previously listed.
--	--

REVIEWER	Ulla Sovio University of Cambridge, United Kingdom
REVIEW RETURNED	29-Apr-2019

GENERAL COMMENTS	The manuscript explored the association between overweight, obesity and adverse pregnancy outcomes in a large, retrospective database from Scotland. Some of the women have contributed more than one delivery to the dataset but this was unaccounted for. Possible solution: If information linking pregnancies from the same women exists, the authors could report the number of women contributing one, two, three or more deliveries, and allow for the correlation structure using appropriate methodology (generalised mixed models). If information does not exist, the authors could restrict the analysis to women with a first pregnancy. Women with a second pregnancy could be reported separately, but the two analyses would be dependent to an extent. The staged approach is problematic too as it leads to selection bias towards the healthiest of the healthy women. The proportion of obese women drops gradually from over 20% to less than 15%. The 15% may be a biased subsample of the 20%, potentially excluding some of the most severely obese women. Excluding women with complicated pregnancies gradually from the analysis has the same problem as adjusting for variables on the causal pathway to the most severe outcomes. The authors then get biased results suggesting e.g. a lower risk of SGA combined with PTB in obese women. However, obese women are at a higher risk of complications which lead to SGA & PTB, and the inverse result reflects a biased sample (the healthiest women). The authors should consider appropriate methods to analyse their data, see e.g. Lange et al, AJE 2013 https://academic.oup.com/aje/article/179/4/513/128034.
--

	The completeness of the recording of BMI increased during the study period (2008 to 2015) from 69% to 98%. Using data from the earlier years when the BMI was missing more often may bias the study sample in case BMI was not missing at random, for example if it was missing more often from the obese women. The authors could comment on this and add any information if available. In addition to odds ratios, the authors should also present raw numbers (%) of cases by exposure status.
--	---

REVIEWER	Dr Michael Fleming University of Glasgow United Kingdom
REVIEW RETURNED	09-May-2019

GENERAL COMMENTS	Thank you for submitting this manuscript. This is an interesting, important and potentially publishable paper however there are several issues which need to be addressed. Comments are below. I think the methods need to be revisited as I am not convinced that a staged analysis is the most appropriate way to look at these outcomes. Simply adjusting for confounders, or undertaking mediation analyses may be more appropriate. The outcomes also need to be defined more clearly throughout the paper. See my comments in full. Reviewer Comments ABSTRACT 1. In the objective (lines 6-8) the authors use the phrase "independent impact of high maternal weight on..." however this should be rephrased to "association between high maternal weight and..." or something similar. Without adjusting for all potential confounders and mediators (including unobserved ones) it is incorrect to suggest that you can accurately model the 'independent' impact of maternal weight on these outcomes. Please also remove/reword the phrase 'independent impact throughout' the paper 2. If possible, word count permitting, it would be nice to see the exact outcomes listed in outcomes measures (line 25-28) 3. Minor comment - throughout the paper you use the phrase "women with overweight and obese" which sounds grammatically incorrect. Perhaps 'obese and overweight women' sounds better STRENGTHS AND LIMITATIONS 4. Lines 18-21 - I disagree that the staged analysis ensures that the independent impact of high maternal weight can be estimated because there still remains unadjusted confounders and mediators (many of which will be unobserved). The only way you could measure the direct effect is if you adjusted for all of these. You haven't in this paper therefore I think independent impact needs to be replaced by association or something similar. 5. Lines 23-28. You have the data to calculate BMI so why did you not further classify obesity into morbid obesity or obesity class II and III given you have cited this as a limitation? I think the paper would benefit from these additional classifications. If it is not possible to do this then you need to state why these categories were not calculated INTRODUCTION
--

6. I'd like to see the intro expanded slightly

- a. to discuss a little more in depth the outcomes that are being measured and why they are specifically important for the future health of mother and child (refs 1-6).
- b. to discuss in more detail the importance to the wider community and NHS as regards the economic consequences and burden - 2 of the 3 refs provided on line 17 are qualitative reviews - are there any bigger more quantitative studies out there?
- c. lines 24-28 - can you please add a reference for NICU admissions and healthcare cost implications
- d. some additional references I think are needed to put this research in context globally. What has been done in other countries and does this study add anything in addition to it being the first to look at these outcomes using Scottish data? Also you say that no 'recent' study in Scotland has looked at these outcomes. Please clarify if any have indeed been undertaken even if small - refer to and discuss previous ones including gaps remaining

METHODS

7. Lines 22-29 - you will also have had to apply to PBPP so this should be included in this section

8. Exposure variables (lines 46-53) - the percentage missing was greater in earlier years - do the authors feel this would have had any bearing on the findings? How will this have affected the validity? This should be discussed as a limitation in the discussion section. Did you contemplate any steps to deal with missing data e.g. imputation? If not why not? Did you consider limiting the study to 2011-2015 when completeness of BMI was better (87%)?

9. Lines 5-16. See earlier comment. Why did you only classify into three groups and not additionally classify morbid obesity? Also can you include a reference for these BMI definitions?

OUTCOMES

10. Lines 26-29. A sentence or two defining pre-eclampsia, placenta praevia, placental abruption and postpartum haemorrhage would be very useful for the non-clinical reader.

11. Lines 26-51. The outcomes need to be much better defined. You have previously mentioned that you are using data from SMR02, SMR01 and SBR but give no more information. Which datasets do each of these outcomes come from? Some will come from SMR02 whilst others will be from SMR01 or SBR. Some may use a combination of both. Did you use ICD-10 codes to capture any of these specific outcomes? If so please give code details. Did you look at all congenital anomalies or just include specific types? Historically congenital anomalies have been ascertained via linkage techniques using SMR02, SMR01 and SBR. What method did you use to ascertain anomalies? More generally why were these specific outcome measures chosen? Was it based on the outcomes you had available from across the datasets or were they chosen for reasons other than that? Are these outcomes all well recorded across the time period? What is the percentage of missing data for each outcome? can this be included somewhere in the results? Why were Apgar (<7) LGA and SGA chosen the way they were and not defined in another way? The growth curve references are 25-30 years old - are there more up to date references available? Stillbirths and neonatal deaths are specifically recorded on the stillbirth and infant death register and by NRS both of which can be linked to SMR02 but which you didn't use in your study. I assume you captured stillbirths and neonatal

deaths on SMR02 - if so what proportion of all stillbirths and neonatal deaths are captured on SMR02? What was the length of follow up for NICU admission?

COVARIATES

12. Lines 3-9. Why did you not also adjust for ethnicity? This is surely an important confounder? What version of SIMD did you use?

13. Lines 3-9. As far as I can see there are additional confounders that haven't been adjusted for here which can affect the outcomes. Just a couple of examples are:

a. time of birth (Pasupathy D et al. Time of birth and risk of neonatal death at term: retrospective cohort study. *BMJ*. 2010;341.)

b. previous caesarian sections, previous stillbirth etc (Oliver-Williams C et al. Previous miscarriage and the subsequent risk of preterm birth in Scotland, 1980–2008: a historical cohort study. *BJOG: An International Journal of Obstetrics & Gynaecology*. 2015;122(11):1525-1534. and Moraitis AA et al. Previous caesarean delivery and the risk of unexplained stillbirth: retrospective cohort study and meta-analysis. *BJOG: An International Journal of Obstetrics & Gynaecology*. 2015;122(11):1467-1474.)

I think you need more discussion on why these confounders were chosen and the limitations of not including any remaining ones. Which other ones are important here and could they be included? If they can't be included discuss their omission in the limitations.

14. Table 2 - it might be beneficial to merge the last two categories of maternal age into 40-49 years. Particularly given that the footnotes in table 3 show that, for several of the outcomes, no cases are encountered in the 45-49 group. Minor comment - deprivation is mis-spelled in the footnotes of table 2 Also is deprivation the correct way round? In SIMD I am used to seeing least deprived as cat 5 and most deprived as cat 1. You have the opposite

DATA ANALYSIS

15. Why did you opt for a staged analyses and not just adjust for stage one conditions when looking at stage 2 outcomes and so on (see Scott-Pillai et al 2013) - rather than omitting women with stage one conditions etc. Omitting women loses lots of valuable cases and reduces sample size (particularly evident when you get towards the latter stage 5 and 6 outcomes) which has resulted in non significant results potentially due to loss of power e.g. congenital anomalies. You could also have adjusted for mediators and confounders using more robust causal mediation analyses which would have given a truer estimate of the direct effect. Were either of these considered? I think either would have been better than staged analyses. If you proceed with staged analyses however I think you need to think carefully about the limitations of doing this and also discuss the strengths and weaknesses of that approach and potential alternatives. Are there any references for using staged analyses. Have other similar studies used this approach? I'd like to see a couple of references to back up this method. Can you do any sensitivity analyses perhaps by adjusting for variables in the usual way (rather than omitting) and comparing the effect sizes? I think the methods need to be revisited

RESULTS

	16. Line 38 - Please just state the actual effect size rather than using language like 'almost 9 fold' - correct other occurrences of this too 17. It would be helpful to get an idea of the absolute rates of some of these outcomes in the population - to put it into context many will be very rare e.g. stillbirths, congenital anomalies etc. An additional supplementary table or some additional text highlighting the rates of occurrence in the different exposure groups could be included and this would help orient the reader 18. TABLE 3 a. Are you not better collapsing the age category to <19 and >40 given you have no cases in the 45-49 age group? b. Why did you additionally look at small for gestational age and preterm as a combined outcome? c. The sample size doesn't make sense for large gestational age - why is this so big compared to the other outcomes in stage 5? DISCUSSION/LIMITATIONS/CONCLUSION 19. Changes are needed to incorporate the comments above REFERENCES 20. some of these e.g, references 12 and 13 are 25-30 years old - are there more up to date references available? ADDITIONAL COMMENT There are some grammatical errors throughout the manuscript - please adjust these accordingly
--	--

VERSION 1 – AUTHOR RESPONSE

Reviewers' comments	Response
Editorial requests	
Please include the study design and setting in the title	We have included the study design and the setting in the title. It is now: a cohort study of high maternal body mass index and the risk of adverse pregnancy, delivery and neonatal outcomes in Scotland.
Please complete and include a STROBE checklist, ensuring that all points are included and state the page numbers where each item can be found.	We have included a STROBE checklist, stating the page number where each item could be found.
Reviewer 1	
Cannot determine if the increase in diabetes (or other outcomes) are the cause of rather than the effect of the reported conclusions.	We are not sure what the reviewer meant, but the focus of this study was on looking at the association between maternal weight status and perinatal risks. Since maternal BMI was based on weight at first antenatal booking, thus preceding delivery by (typically) some months, there is no issue here of reverse causation – i.e. the perinatal outcomes causing the maternal BMI.
Reviewer 2	
Page 4 introduction: makes it more likely that a.....	We have addressed this.
Page 4: another reference could be added to the fifth line Aune D, Saugstad OD, Henriksen T, Tonstad S.	We have added this useful reference - thanks.

Maternal body mass index and the risk of fetal death, stillbirth, and infant death: a systematic review and meta-analysis. JAMA. 2014 Apr 16;311(15):1536-46.	
Table 3. Please add number of cases in total at the left as well as for each category of BMI for each outcome. That would be more informative than the percentages that are provided. Give some more info of what is the comparison for Apgar score.	We have added the number (and percentages) of cases in total for the controls as well as for each category of BMI for each outcome, including Apgar score. We have also attached a web-appendix of the data fields we received with the dataset and this provides useful source of information for definitions.
Do you have data regarding very preterm births (before week 32)? Do you have data on macrosomia, infant death, shoulder dystocia, intrauterine growth restriction, neonatal jaundice, neonatal hypoglycemia, resuscitation, wound infection, urinary tract infection, and/or maternal mortality? Because of the limited available data on these outcomes it would have been good to see more data on these outcomes. Would have been great if it would be possible to add results for these outcomes.	We agree that it would have been good to see data on the outcomes listed by the Reviewer, but unfortunately the administrative dataset we received did not include these outcomes. All relevant outcome data we received were used in the analysis.
It would have been good if the authors could add analyses for more detailed categories of obesity including grade 1 (30-<35), 2 (35-<40) and 3 (40+) in a sensitivity analysis perhaps in a supplementary table. And perhaps also add results using more refined categories of weight BMI <18.5, 18.5-<20.0, 20.0-<22.5, 22.5-<25.0, 25.0-<27.5, 27.5-<30.0, 30.0-<32.5, 32.5-<35.0, 35.0-<40, 40+. The study is very large (and the number of cases must be huge as well) - especially when looking at the ORs and CIs in table 3. Most previous studies have only used the WHO classification of weight, but using more refined categories can be important in determining the optimal level of BMI in pregnancy. Therefore it would be important to provide results with more detailed categories.	Unfortunately the administrative dataset we received from the data custodian included only the following categories of BMI: health weight/underweight, overweight and obesity categories. This means it is not possible to look at detailed categories of obesity. Also, our revised analysis shows that for several outcomes reduced obesity cell sizes make the analysis less than optimally powerful. This means that even if we had finely graded BMI categories, they would certainly lack power for these outcomes.
Page 11, 3rd paragraph: Being overweight or obese significantly decreased the risk of low Apgar score.....	We have addressed this.
Page 12: it says that most previous studies have focused on obese women. I'm not so sure if this is a good point or even true as there are a large number of studies that have analysed both overweight and obesity. Same paragraph: women are likely to be overweight rather than obese, It could also be argued that many women are on the high end of the normal weight range and it is unknown what is the optimal BMI. Clarifying if there is increased risk even within the high-normal BMI range might move the existing literature even further and contribute to better weight recommendations for pregnant women.	We have amended the paragraph and the sentence now reads: Aside from obesity, we also examined overweight because in most populations, a greater number of women are overweight rather than obese, so it is important to also understand the impact of overweight on pregnancy and neonatal outcomes.
Page 14, Discussion, paragraph 4: The lack of association between overweight and stillbirth is almost certainly because of lack of power (OR 1.24 (0.94-1.64) so this should be mentioned as a possible explanation here. Also, even if some of the previous studies have varied in the results for overweight the meta-analysis below detected a clear association between increasing weight even from high normal BMI, to overweight and obesity. Aune D, Saugstad OD, Henriksen T, Tonstad S.	The revised analysis now shows that overweight is associated with stillbirth in line with Aune et al. 2014 systematic review. We have revised the paragraph accordingly.

Maternal body mass index and the risk of fetal death, stillbirth, and infant death: a systematic review and meta-analysis. JAMA. 2014 Apr 16;311(15):1536-46.	
Page 14, paragraph 5 on overweight/obesity and risk of SGA and preterm birth. The inverse association with SGA has been found in many other studies, and should not be considered unexpected as adiposity increases risk of LGA and macrosomia. For preterm birth the results have been mixed with positive, null and inverse association reported, however, if you look at studies on very preterm birth (<34 or <32 weeks) the results are more consistent with an increased risk. For this reason I asked in one of my previous comments if you could also add results for very preterm births.	In the revised analysis, overweight/obesity and risk of SGA and preterm birth were no longer significant. However, overweight/obesity and risk of LGA were significant. We have amended the paragraph accordingly.
One thing I am wondering about is if the authors by using the staged approach (sequentially excluding subjects with the previously listed conditions when analyzing each outcome) are introducing some sort of bias or selection effect to the results or that the results may become less and less generalizable the further down the table you get. For example maternal adiposity increases the risk of gestational hypertension/preeclampsia and gestational diabetes which all may contribute to the increased risk of stillbirth observed with maternal adiposity. I wonder if the reported associations therefore might be more modest (although still quite strong) than if the full population was used. Are the results very different if using the full study population without making such exclusions?	We recognise this issue, which was also raised by a number of the other reviewers. However, we haven't been able to find an analytical technique which could accommodate the list of multiple perinatal outcomes we were able to examine which is a novel aspect of this paper. The data available to us does not allow us to identify the cases for example of gestational diabetes which are attributable to the mothers weight, and those which are independent of the mother's weight, to then examine the impact of the mother's weight status on subsequent outcomes. Mediation analysis would only allow the exploration of a handful of our many outcomes.
Ref. 7: Heslehurst	We have removed the "s" at the end of the name.
Reviewer 3	
1. The amount of outcomes limit the interpretation of the study and the scope of the results	This study like any other research study has weaknesses. We believe that the outcomes we examine are appropriate to scope of the study and the size of the dataset. Indeed, a classical advantage of cohort studies such as ours is their ability to study many outcomes simultaneously. This is especially relevant when almost all of them are associated with one clinical characteristic: maternal weight status at first obstetrical booking.
2. Important confounding factors are not included in the analysis and therefore make the results inconclusive or imprecise : such as : - Previous diabetes - Chronic Hypertension - Previous bariatric surgery - Weight gain during pregnancy - For multiparous women with previous hypertension or Preeclampsia, use of prophylactic aspirin - Spontaneous or ART pregnancy - Types of Fetal abnormalities	We agree that there may be other confounding factors that were not included in the analysis but unfortunately the administrative dataset we received did not include these variables. We have acknowledged this in the limitations section of the paper. We have limited the analysis to first time pregnancies. All relevant variables we received were used in the analysis.
3. Uncompleted data about BMI, the gestational age of the first consultation	As we explained in the methods, over 80% of pregnant women in Scotland present themselves for antenatal care before 12 weeks of pregnancy, where height and weight (BMI) are measured by the midwife. The gestational age of the first consultation may therefore vary - we unfortunately do not have access to this information.

4. Sub-class of obesity are not differentiated	Unfortunately the administrative dataset we received from the data custodian included only the following categories of BMI: health weight/underweight, overweight and obesity categories. This means it is not possible to look at detailed categories of obesity. Also, our revised analysis shows that for several outcomes reduced obesity cell sizes make the analysis less than optimally powerful. This means that even if we had finely graded BMI categories, they would certainly lack power for these outcomes.
5. As written by the authors, women could be included more than one time if they had more than one birth.	We have revised the analysis and limited it to not only singleton pregnancies, but also first pregnancies to ensure that the cases in the sample are more independent.
6. Mode of screening for gestational diabetes should be described	We have now attached a web-appendix of the data fields we received with the dataset and this provides useful source of information for definitions.
7. For each outcomes, prevalence should be given in the group control	We have addressed this in Table 3.
8. For condition occurring during pregnancy, antepartum or post partum diagnosis should be precised.	We have now attached a web-appendix of the data fields we received with the dataset and this provides useful source of information for definitions.
9. The stage fashion according to the clinical timelines is not accurate: Stage 3 is dependant of stage 5 (for example, induction of labor is more frequent in case of SGA...)	We have revised the staged analysis approach (see figure 1) and we now believe that this has now addressed the reviewer's point.
10. Induction of labor should be described (reasons). Emergency caesarean section should be separated between planned cesarean section and cesarean section during trial of labor.	We have now attached a web-appendix of the data fields we received with the dataset and this provides useful source of information for definitions, data sources and how outcomes were defined. Emergency Caesarean section was differentiated from elective (planned) Caesarean section in the data we received.
11. Collider and confounding factors in such analysis could be described with a directed acyclic graph It would be more appropriate to address the issue of one only one outcomes so that to identify all factors that could contribute to the result. One question for one response in nulliparous only. Confounding bias, collider stratification, and selection biases, are important when addressing issues in obstetrics and perinatal epidemiology. Therefore when reading this paper, results cannot be used in daily practice.	If conducting this study from scratch this would be an excellent idea to use a Directed Acyclic Graph in order to describe these complex relationships and refine analytical strategies, and one that we would consider when conducting similar work. However, to create a suitably complex DAG for so many interdependent perinatal outcomes, and their potential pre-natal risk factors, confounders and mediators/modifiers, using a mixed clinical and epidemiological team, would surely be a research project on its own. Whilst we can understand why it is useful for clinicians to have a separate answer to each outcome, this epidemiological paper seeks to assess outcomes in the context of the clinical environment – in this case antenatal and perinatal care - in which they happen. We feel that this paper is still useful to clinicians when considering the many, often inter-related, outcomes of women with overweight or obesity. We have however restricted the analyses to not only singleton pregnancies, but also first pregnancies to

	ensure that the sample cases are more independent.
Association between stillbirth and obesity is not proven	Other studies, including a review that reviewer 2 suggested, all show that maternal obesity is associated with stillbirth; so we believe the statement in paragraph one of the introduction we made concerning this should still stand.
Reference 2 is not appropriate	Sorry for the oversight. We have now removed this reference.
The decreased risk of SGA and preterm delivery with overweight could be the result of collider factors. Protective effects cannot be explained by this study. This paradox may often reflects a statistical problem	The revised analysis shows that this association is no longer significant.
This paper could be submitted again as a new submission with: a. restricting analysis to one outcome such as : risk of preeclampsia in overweight nulliparous women (it is an example) b. Caution should be given to confounding factors when addressing issue in peri-natality c. Without missing data - With data on previous maternal condition as previously listed.	a. The scope of our analysis – the excess risk of the full panoply of perinatal complications associated with excessive maternal weight at first booking -- means that restricting the analysis to only one outcome will not serve the purpose of the paper. b. We have carefully considered all relevant confounding factors we received from this administrative data. c. Unfortunately exact date of delivery and data on previous maternal condition were not part of the administrative data we received so it is difficult to consider these in the analysis.
Reviewer 4	
Some of the women have contributed more than one delivery to the dataset but this is was unaccounted for. Possible solution: If information linking pregnancies from the same women exists, the authors could report the number of women contributing one, two, three or more deliveries, and allow for the correlation structure using appropriate methodology (generalised mixed models). If information does not exist, the authors could restrict the analysis to women with a first pregnancy. Women with a second pregnancy could be reported separately, but the two analyses would be dependent to an extent.	We have revised the analysis and limited it to not only singleton pregnancies, but and also first pregnancies to ensure that the sample is more independent.
The staged approach is problematic too as it leads to selection bias towards the healthiest of the healthy women. The proportion of obese women drops gradually from over 20% to less than 15%. The 15% may be a biased subsample of the 20%, potentially excluding some of the most severely obese women. Excluding women with complicated pregnancies gradually from the analysis has the same problem as adjusting for variables on the causal pathway to the most severe outcomes. The authors then get biased results suggesting e.g. a lower risk of SGA combined with PTB in obese women. However, obese women are at a higher risk of complications which lead to SGA & PTB, and the inverse result reflects a biased sample (the healthiest women). The authors should consider	Thank you for recommending paper by Lange et al. which we have reviewed. We agree that mediation analysis would be a better analytical approach when examining a small number of outcomes. However, we were not able to identify a published analytical approach (including those involving DAGs) which could accommodate the list of multiple, inter-related perinatal outcomes we considered. Overweight or obesity are not the necessary cause of any of the perinatal outcomes examined. Consequently, the staged approach purposefully biases towards the healthier pregnancies in order to minimise the chance of confounding the risk of overweight with the risk of the previous perinatal outcomes.

appropriate methods to analyse their data, see e.g. Lange et al, AJE 2013 https://academic.oup.com/aje/article/179/4/513/128034 .	
The completeness of the recording of BMI increased during the study period (2008 to 2015) from 69% to 98%. Using data from the earlier years when the BMI was missing more often may bias the study sample in case BMI was not missing at random, for example if it was missing more often from the obese women. The authors could comment on this and add any information if available.	Unfortunately the administrative dataset we received did not include date of delivery so it is difficult to know the BMI distribution across the study period. We have now acknowledged this as a limitation of the study.
In addition to odds ratios, the authors should also present raw numbers (%) of cases by exposure status.	We have addressed this in table 3.
Reviewer 5	
ABSTRACT 1. In the objective (lines 6-8) the authors use the phrase "independent impact of high maternal weight on..." however this should be rephrased to "association between high maternal weight and...." or something similar. Without adjusting for all potential confounders and mediators (including unobserved ones) it is incorrect to suggest that you can accurately model the 'independent' impact of maternal weight on these outcomes. Please also remove/reword the phrase independent 'impact throughout' the paper	We have now removed the phrase "independent impact" throughout the paper.
2. If possible, word count permitting, it would be nice to see the exact outcomes listed in outcomes measures (line 25-28)	We have listed all the outcomes we studied in the abstract.
3. Minor comment - throughout the paper you use the phrase "women with overweight and obese" which sounds grammatically incorrect. Perhaps 'obese and overweight women' sounds better	We appreciate that this is a little clunky, however it is a journal requirement that 'people-first' language is used.
STRENGTHS AND LIMITATIONS 4. Lines 18-21 - I disagree that the staged analysis ensures that the independent impact of high maternal weight can be estimated because there still remains unadjusted confounders and mediators (many of which will be unobserved). The only way you could measure the direct effect is if you adjusted for all of these. You haven't in this paper therefore I think independent impact needs to be replaced by association or something similar.	We have now removed the phrase "independent impact" throughout the paper.
5. Lines 23-28. You have the data to calculate BMI so why did you not further classify obesity into morbid obesity or obesity class II and III given you have cited this as a limitation? I think the paper would benefit from these additional classifications. If it is not possible to do this then you need to state why these categories were not calculated	Unfortunately the administrative dataset we received from the data custodian included only the following categories of BMI: health weight/underweight, overweight and obesity categories. This means it is not possible to look at detailed categories of obesity. Also, our revised analysis shows that for several outcomes reduced obesity cell sizes make the analysis less than optimally powerful. This means that even if we had finely graded BMI categories, they would certainly lack power for these outcomes.
INTRODUCTION 6. I'd like to see the intro expanded slightly a. to discuss a little more in depth the outcomes that are	a. We have expanded on the introduction. We have discussed more of the outcomes being measured.

being measured and why they are specifically important for the future health of mother and child (refs 1-6). b. to discuss in more detail the importance to the wider community and NHS as regards the economic consequences and burden - 2 of the 3 refs provided on line 17 are qualitative reviews - are there any bigger more quantitative studies out there? c. lines 24-28 - can you please add a reference for NICU admissions and healthcare cost implications d. some additional references I think are needed to put this research in context globally. What has been done in other countries and does this study add anything in addition to it being the first to look at these outcomes using Scottish data? Also you say that no 'recent' study in Scotland has looked at these outcomes. Please clarify if any have indeed been undertaken even if small - refer to and discuss previous ones including gaps remaining	b. We have added references to quantitative studies looking at economic consequences and burden of maternal obesity. c. We have addressed this. d. We have added additional references to capture the international as well as the Scottish context of the study.
METHODS 7. Lines 22-29 - you will also have had to apply to PBPP so this should be included in this section	Electronic Data Research and Innovation Service of the Information Service Division in Scotland reviewed our proposal and confirmed that PBPP was not required, in that the data we requested was categorised by ISD staff as a “confidential data release”.
8. Exposure variables (lines 46-53) - the percentage missing was greater in earlier years - do the authors feel this would have had any bearing on the findings? How will this have affected the validity? This should be discussed as a limitation in the discussion section. Did you contemplate any steps to deal with missing data e.g. imputation? If not why not? Did you consider limiting the study to 2011-2015 when completeness of BMI was better (87%)?	Unfortunately the administrative dataset we received did not include date of delivery, so it is difficult to restrict the analysis to 2011-2015. We have now acknowledged this as a limitation of the study. We did not consider imputation, as we wanted the study to be based on real data as much as possible.
9. Lines 5-16. See earlier comment. Why did you only classify into three groups and not additionally classify morbid obesity? Also can you include a reference for these BMI definitions?	Unfortunately the administrative dataset we received from the data custodian included only the following categories of BMI: health weight/underweight, overweight and obesity categories. This means it is not possible to look at detailed categories of obesity. Also, our revised analysis shows that for several outcomes reduced obesity cell sizes make the analysis less than optimally powerful. This means that even if we had finely graded BMI categories, they would certainly lack power for these outcomes. We have limited the analysis to adults, so the 1990 growth curve references are no longer necessary to define BMI categories.
OUTCOMES 10. Lines 26-29. A sentence or two defining pre-eclampsia, placenta praevia, placental abruption and postpartum haemorrhage would be very useful for the non-clinical reader. What was the length of follow up for NICU admission?	We have now defined these terms.
11. Lines 26-51. The outcomes need to be much better defined. You have previously mentioned that you are using data from SMR02, SMR01 and SBR but give no more information. Which datasets do each of these outcomes come from? Some will come from SMR02	We have now attached a web-appendix of the data fields we received with the dataset and this usefully provides “source of information” for definitions, data sources, in terms of how outcomes were defined.

whilst others will be from SMR01 or SBR. Some may use a combination of both. Did you use ICD-10 codes to capture any of these specific outcomes? If so please give code details. Did you look at all congenital anomalies or just include specific types? Historically congenital anomalies have been ascertained via linkage techniques using SMR02, SMR01 and SBR. What method did you use to ascertain anomalies? More generally why were these specific outcome measures chosen? Was it based on the outcomes you had available from across the datasets or were they chosen for reasons other than that? Are these outcomes all well recorded across the time period? What is the percentage of missing data for each outcome? can this be included somewhere in the results? Why were Apgar (<7) LGA and SGA chosen the way they were and not defined in another way? The growth curve references are 25-30 years old - are there more up to date references available? Stillbirths and neonatal deaths are specifically recorded on the stillbirth and infant death register and by NRS both of which can be linked to SMR02 but which you didn't use in your study. I assume you captured stillbirths and neonatal deaths on SMR02 - if so what proportion of all stillbirths and neonatal deaths are captured on SMR02?	
COVARIATES 12. Lines 3-9. Why did you not also adjust for ethnicity? This is surely an important confounder? What version of SIMD did you use?	Ethnicity is a special category data and therefore more sensitive than could be released within a confidential data release. We would also point out that non-white/British subpopulations in Scotland are typically so small that analyses by such categories typically lack power, even for commoner outcomes. We used the Carstairs 2001 SIMD quintiles for Scotland. We have made this explicit in Table 2.
13. Lines 3-9. As far as I can see there are additional confounders that haven't been adjusted for here which can affect the outcomes. Just a couple of examples are: a. time of birth (Pasupathy D et al. Time of birth and risk of neonatal death at term: retrospective cohort study. BMJ. 2010;341.) b. previous caesarian sections, previous stillbirth etc (Oliver-Williams C et al. Previous miscarriage and the subsequent risk of preterm birth in Scotland, 1980–2008: a historical cohort study. BJOG: An International Journal of Obstetrics & Gynaecology. 2015;122(11):1525-1534. and Moraitis AA et al. Previous caesarean delivery and the risk of unexplained stillbirth: retrospective cohort study and meta-analysis. BJOG: An International Journal of Obstetrics & Gynaecology. 2015;122(11):1467-1474.) I think you need more discussion on why these confounders were chosen and the limitations of not including any remaining ones. Which other ones are important here and could they be included? If they can't be included discuss their omission in the limitations.	We agree that there may be other confounding factors that were not included in the analysis but unfortunately the administrative dataset we received did not include these variables. We have acknowledged this in the limitations section of the paper. All relevant variables we received were used in the analysis.
14. Table 2 - it might be beneficial to merge the last two categories of maternal age into 40-49 years. Particularly given that the footnotes in table 3 show that, for several of the outcomes, no cases are encountered in the 45-49	We have limited the age range to 20-40 to address the issue of small numbers of cases within age groups.

group. Minor comment - deprivation is mis-spelled in the footnotes of table 2 Also is deprivation the correct way round? In SIMD I am used to seeing least deprived as cat 5 and most deprived as cat 1. You have the opposite	Sorry for the oversight, we have now provided correct spelling for deprivation in Table 2. Yes, we can confirm from the data fields received from eDRIS that deprivation is in correct order – we note that that order as found in Scottish Government/NHS publications, has been reversed in recent years, as compared to the past.
DATA ANALYSIS 15. Why did you opt for a staged analyses and not just adjust for stage one conditions when looking at stage 2 outcomes and so on (see Scott-Pillai et al 2013) - rather than omitting women with stage one conditions etc. Omitting women loses lots of valuable cases and reduces sample size (particularly evident when you get towards the latter stage 5 and 6 outcomes) which has resulted in non significant results potentially due to loss of power e.g. congenital anomalies. You could also have adjusted for mediators and confounders using more robust causal mediation analyses which would have given a truer estimate of the direct effect. Were either of these considered? I think either would have been better than staged analyses. If you proceed with staged analyses however I think you need to think carefully about the limitations of doing this and also discuss the strengths and weaknesses of that approach and potential alternatives. Are there any references for using staged analyses. Have other similar studies used this approach? I'd like to see a couple of references to back up this method. Can you do any sensitivity analyses perhaps by adjusting for variables in the usual way (rather than omitting) and comparing the effect sizes? I think the methods need to be revisited	We selected the staged approach in order to address the issue that you have identified around the compounding risk of perinatal outcomes. As described above, maternal overweight or obesity is not the necessary cause of any of the perinatal outcomes examined, and the data do not allow us to distinguish between cases attributable to overweight or obesity and those which are not. Mediation analysis would have given information about the proportion of cases attributable to overweight or obesity, but not which cases. Therefore, mediation analysis would not have enabled the estimation of the associations which interested us. Furthermore, none of the mediation analysis techniques seem to be able to handle the large number of outcomes we wanted to examine (see our argument for clinical relevance above). We have amended the language throughout the paper and hopefully provide a better reflection of the interpretation of the findings and the limitations of the approach. We were not able to find any studies which had used the same approach.
RESULTS 16. Line 38 - Please just state the actual effect size rather than using language like 'almost 9 fold' - correct other occurrences of this too.	Done.
17. It would be helpful to get an idea of the absolute rates of some of these outcomes in the population - to put it into context many will be very rare e.g. stillbirths, congenital anomalies etc. An additional supplementary table or some additional text highlighting the rates of occurrence in the different exposure groups could be included and this would help orient the reader available?	We have now addressed this in Table 3.
18. TABLE 3 a. Are you not better collapsing the age category to <19 and >40 given you have no cases in the 45-49 age group? b. Why did you additionally look at small for gestational age and preterm as a combined outcome? c. The sample size doesn't make sense for large gestational age - why is this so big compared to the other outcomes in stage 5?	a. We have now addressed this issue in Table 3. b. Additive effect of SGA and preterm evidence in outcomes e.g.: Leviton, A., Fichorova, R. N., O'Shea, T. M., Kuban, K., Paneth, N., Dammann, O., & Allred, E. N. (2013). Two-hit model of brain damage in the very preterm newborn: small for gestational age and postnatal systemic inflammation. Pediatric research, 73(3), 362. Giapros, V., Drougia, A., Krallis, N., Theocharis, P., & Andronikou, S. (2012). Morbidity and mortality patterns in small-for-gestational age infants born

	preterm. The Journal of Maternal-Fetal & Neonatal Medicine, 25(2), 153-157. c. We have checked the data again and can confirm that there are a lot more cases of LGA than any of the other similar outcomes.
DISCUSSION/LIMITATIONS/CONCLUSION 19. Changes are needed to incorporate the comments above	We have made the necessary changes.
REFERENCES 20. some of these e.g, references 12 and 13 are 25-30 years old - are there more up to date references	There are no current references for the UK 1990 BMI reference curves. They both remain the most widely cited growth reference curves, because updating them would mean redefining overweight and obesity in children, in an era after the advent of the current obesity pandemic. We need to use a historical reference population to define changes in the prevalence of overweight or obesity as the rate of natural (pre-pandemic) growth in children means that fixed cut-points can be used as in adults. The methods proposed by Cole – still by far the most widely used -- mean that age and gender specific cut-points are used in children.
ADDITIONAL COMMENT There are some grammatical errors throughout the manuscript - please adjust these accordingly	We have read through the entire paper and corrected all grammatical errors, especially those related to “women with overweight and obesity” as suggested by this Reviewer 5.

VERSION 2 – REVIEW

REVIEWER	Dagfinn Aune Imperial College London, UK
REVIEW RETURNED	22-Jul-2019

GENERAL COMMENTS	The authors have responded to most of my comments, however, I'm still a bit unsure about whether the staged approach to the analysis is appropriate. I wonder if it would better to include all subjects across all analyses without stage-wise exclusions? Gestational hypertension, diabetes and preeclampsia may be on the biological pathway from overweight/obesity to stillbirth and removing those cases from the analysis may lead to an underestimate of the association between overweight/obesity and stillbirth for example. I think in the current approach you may be getting sequentially more and more biased results the more pregnancy conditions you exclude. I would have preferred to see analyses of the full dataset across all outcomes. That is how previous studies have analysed their data. At least you could do this and report it as a sensitivity analysis in the online supplement. See for example: Ovesen P, Rasmussen S, Kesmodel U. Effect of prepregnancy maternal overweight and obesity on pregnancy outcome. Obstet Gynecol. 2011 Aug;118(2 Pt 1):305-12. Syngelaki A, Bredaki FE, Vaikousi E, Maiz N, Nicolaidis KH. Body mass index at 11-13 weeks' gestation and pregnancy complications. Fetal Diagn Ther. 2011;30(4):250-65.
---

	Raatikainen K, Heiskanen N, Heinonen S. Transition from overweight to obesity worsens pregnancy outcome in a BMI-dependent manner. Obesity (Silver Spring). 2006 Jan;14(1):165-71. Page 10: lower CI = OR = 8.71. Please correct this. Page 12: risk of stillbirth was not significantly higher - DELETE not Page 13, last line: women of normal weight..... separate of normal
--	--

REVIEWER	Michael Fleming University of Glasgow, Glasgow, Scotland, UK
REVIEW RETURNED	11-Jul-2019

GENERAL COMMENTS	Thank you for this resubmitted manuscript. Whilst this manuscript has improved I still have major concerns about the statistical analyses employed. 1. I do not feel that the authors have adequately allayed my concerns around the staged analyses and I see that some of the other reviewers have raised the same concerns. I do not feel that these results are generalisable because of the selection bias that occurs due to omitting women who experience some of the earlier staged outcomes. The results will be biased towards the healthier (and likely less obese) women. The authors have commented that this has deliberately been done however it remains that the effect sizes are only relevant to the specific population of women represented following all of the various omissions. Other reviewers have correctly commented that it is important to understand how obesity is associated with each of the outcomes within the full cohort rather than a manufactured (and as it happens biased) subset. The authors have again commented that the analyses demonstrate the 'precise impact of high maternal weight on each outcome'. This is untrue in my opinion. Rather these analyses at best show the association between maternal weight and the outcomes with several caveats applied (the omissions). I previously queried whether the authors can rerun a sensitivity analyses to compare the effects with what would happen if the whole population is used. I believe another reviewer also asked for this and I still think that this would be very beneficial (and required). It is unfortunate that the authors could not find any evidence of this technique being used anywhere else in the literature because, as stated previously I have major doubts that this is the correct method to use. The lack of other studies using this method strengthens my view that these are not the correct analyses to employ. I take on board the authors viewpoint that mediation analyses may not be appropriate because of the high number of outcomes. However I certainly think that analyses using the whole population with adequate adjustment for confounders would be useful at least as a reference point. Another option, as pointed out by one of the other reviewers, would be to reduce the number of outcomes and employ mediation analyses on those. I do feel that the number of outcomes also complicates the paper. 2. I believe that the authors are not getting the most out of the data because of the extract that they have received from eDRIS which doesn't include several key confounders and doesn't include date of birth to enable missing values around BMI to be further investigated. BMI also could not be categorized further because of the nature of the extract. I think some more discussion should be included to mention these limitations and the other confounders
---

	that might be important which have not been considered. The authors state in the limitation section that these limitations are common in studies using administrative data. This is untrue. In fact administrative data can be very powerful however in this instance it seems that the authors had to forego much of the detail in these data to enable them to speed up receipt of the extract with increased anonymization. For this reason the study is not as robust as it could be if more data had been provided including variables such as ethnicity, previous caesarian sections, previous stillbirths, time of birth etc 3. I still have doubts around how well some of the outcomes such as stillbirths and congenital anomalies are recorded given they are routinely recorded over multiple datasets. Given the authors only used SMR02 for stillbirths and SBR for congenital anomalies how well do these datasets pick these up? Also it would be useful for the authors to comment on the completeness of other variables such as NICU admission on SMR02. Can we see some information around the amount of missing data for all of the outcomes?
--	--

VERSION 2 – AUTHOR RESPONSE

Reviewers' comments	Response
Reviewer 5	
1. I do not feel that the authors have adequately allayed my concerns around the staged analyses and I see that some of the other reviewers have raised the same concerns. I do not feel that these results are generalisable because of the selection bias that occurs due to omitting women who experience some of the earlier staged outcomes. The results will be biased towards the healthier (and likely less obese) women. The authors have commented that this has deliberately been done however it remains that the effect sizes are only relevant to the specific population of women represented following all of the various omissions. Other reviewers have correctly commented that it is important to understand how obesity is associated with each of the outcomes within the full cohort rather than a manufactured (and as it happens biased) subset. The authors have again commented that the analyses demonstrate the 'precise impact of high maternal weight on each outcome'. This is untrue in my opinion. Rather these analyses at best show the association between maternal weight and the outcomes with several caveats applied (the omissions). I previously queried whether the authors can rerun a sensitivity analyses to compare the effects with what would happen if the whole population is used. I believe another reviewer also asked for this and I still think that this would be very beneficial (and required). It is unfortunate that the authors could not find any evidence of this technique being used anywhere else in the literature because, as stated previously I have major doubts that this is the correct method to use. The lack of	We have decided against using our original staged analysis approach and have now re-analysed the data based on the recommendations by both reviewers. We have analysed the dataset across all outcomes with adequate adjustment for relevant confounders. We have also reduced the number of outcomes analysed (by not analysing stillbirth, neonatal death, NICU admission, congenital anomaly and postpartum haemorrhage). As noted by this reviewer, these outcomes are not completely ascertained in the dataset we used.

other studies using this method strengthens my view that these are not the correct analyses to employ. I take on board the authors viewpoint that mediation analyses may not be appropriate because of the high number of outcomes. However I certainly think that analyses using the whole population with adequate adjustment for confounders would be useful at least as a reference point. Another option, as pointed out by one of the other reviewers, would be to reduce the number of outcomes and employ mediation analyses on those. I do feel that the number of outcomes also complicates the paper.	
2. I believe that the authors are not getting the most out of the data because of the extract that they have received from eDRIS which doesn't include several key confounders and doesn't include date of birth to enable missing values around BMI to be further investigated. BMI also could not be categorized further because of the nature of the extract. I think some more discussion should be included to mention these limitations and the other confounders that might be important which have not been considered. The authors state in the limitation section that these limitations are common in studies using administrative data. This is untrue. In fact administrative data can be very powerful however in this instance it seems that the authors had to forego much of the detail in these data to enable them to speed up receipt of the extract with increased anonymization. For this reason the study is not as robust as it could be if more data had been provided including variables such as ethnicity, previous caesarian sections, previous stillbirths, time of birth etc	We have elaborated on the reviewer's concern in the limitations section, by stating that some relevant confounders were not available in the dataset which we accessed and so could not be used.
3. I still have doubts around how well some of the outcomes such as stillbirths and congenital anomalies are recorded given they are routinely recorded over multiple datasets. Given the authors only used SMR02 for stillbirths and SBR for congenital anomalies how well do these datasets pick these up? Also it would be useful for the authors to comment on the completeness of other variables such as NICU admission on SMR02. Can we see some information around the amount of missing data for all of the outcomes?	We have decided not to analyse the neonatal outcomes that are not completely ascertained in the dataset we used, as the reviewer noted.
Reviewer 2	
The authors have responded to most of my comments, however, I'm still a bit unsure about whether the staged approach to the analysis is appropriate. I wonder if it would better to include all subjects across all analyses without stage-wise	We have decided against using our original staged analysis approach and have now re-analysed the data based on the recommendation by both reviewers. We have analysed the dataset across all outcomes with adequate adjustment for relevant

exclusions? Gestational hypertension, diabetes and preeclampsia may be on the biological pathway from overweight/obesity to stillbirth and removing those cases from the analysis may lead to an underestimate of the association between overweight/obesity and stillbirth for example. I think in the current approach you may be getting sequentially more and more biased results the more pregnancy conditions you exclude. I would have preferred to see analyses of the full dataset across all outcomes.	confounders. [Strikingly, we note that hardly any substantive change in the estimated ORs resulted from this new analytic approach, compared to our previous staged approach].
Page 10: lower CI = OR = 8.71. Please correct this.	Sorry for the oversight, this has now been addressed.
Page 12: risk of stillbirth was not significantly higher - DELETE not	This sentence has been completely removed from the revised manuscript.
Page 13, last line: women of normal weight..... separate ofnormal	We have addressed this.

VERSION 3 – REVIEW

REVIEWER	Dagfinn Aune Imperial College London
REVIEW RETURNED	27-Aug-2019

GENERAL COMMENTS	The authors have redone the analyses using the full cohort rather than the staged design and this is appropriate. I think it is a pity that some of the results for certain outcomes - stillbirth, postpartum hemorrhage and NICU admission have been omitted for the reason of the outcome assessment being incomplete, as they show similar results as other studies (if anything, an incomplete outcome assessment would most likely have led to underestimation of the observed associations). Other than that I don't have any further comments.
---

REVIEWER	Michael Fleming University of Glasgow Scotland, UK
REVIEW RETURNED	30-Sep-2019

GENERAL COMMENTS	Thank you for this revised manuscript and for addressing my previous comments. I feel that this latest draft is a much-improved paper. Firstly, the authors have removed some of the outcomes which I previously felt could not be adequately studied solely using the datasets in this study due to inadequate case ascertainment (stillbirths, congenital anomalies etc.). As a by-product of removing these outcomes, the paper is now actually easier to follow with fewer outcomes studied. Secondly, the authors have now
---

adequately addressed the limitations – specifically the lack of information on several potentially important confounders. Thirdly, and most importantly, the reviewers have decided against using their initial staged analyses approach. I note that the authors have commented that there was no substantive change in the odds ratios after running the latest analyses and this is reassuring. However, I believe that the current analyses is the most appropriate, in terms of robustness and accuracy, but also in terms of ease of interpretation and wider generalisability. In saying this, I still have some final outstanding comments detailed below which I think need to be addressed.

1. Introduction

My major outstanding comment is that the introduction no longer adequately reflects the paper. The authors focus their introduction on stillbirths, neonatal/perinatal/infant deaths and admissions to neonatal units; however, the authors have removed these outcomes from the manuscript and so they are no longer relevant. Similarly, with the exception of one sentence mentioning diabetes, and hypertension, the introduction does not actually mention any of the other outcomes that *are* studied. The introduction states that no *recent* study in Scotland has investigated the effect of high BMI on pregnancy and delivery outcomes. Does this mean that there are *some* studies in Scotland but none more recently? Or none at all? What does more recently mean? What about studies worldwide? In light of the fact many of the original outcomes have been removed from the paper, I think the authors need to revisit the introduction in order to make it more appropriate and relevant to the rest of the paper. In particular, I think the reader needs to see a little bit more about what previous studies worldwide have found with respect to associations with each of the outcomes.

2. Strengths and limitations box

The authors state that they have used a “national database covering *all major* maternal and neonatal outcomes in Scotland over eight years”. In light of several of the most major outcomes (stillbirths, congenital anomalies etc.) having actually been removed this is no longer true. Please change the wording to something more appropriate.

3. In the next sentence, the authors state “adequate adjustment for confounders”. Again this is not accurate given the study is missing several potentially important confounders. The authors have highlighted this as a limitation and later state that they have “a limited set of confounders”. Please therefore remove adequate from the main box and replace with something more appropriate.

	4. Methods Minor comment/observation - I do not really understand the term “clinical audit” in this context. Surely, this is just a regular retrospective cohort study using routinely collected administrative datasets? 5. Minor comment/suggestion - Table 1 is very small and so perhaps the definitions of BMI could just be described as text within the body of the paper rather than a separate table? 6. It would be informative for the reader to see an additional line in table 2 showing the number and percentage of missing data for each of the covariates. Smoking in particular is renowned as having rather a lot of missing data. How did the authors deal with missing data? Were women with missing data just dropped from the analyses or did you think about imputation? 7. Data analysis The authors state that each outcome was additionally adjusted for conditions that precede or can occur alongside it. I think it would be useful to reflect this within the footnotes of table 3 i.e. can you clarify in the footnotes which variables were used in each of the different analyses? 8. Results In the second paragraph, the sentence beginning “the risk of the three conditions....” should be made clearer for the reader. Could this be rephrased as “the risk of gestational diabetes, preeclampsia and hypertension.....” 9. Furthermore, the authors have not discussed the results for hypertension, pre or post term delivery or induction of labour in the results text meaning that the reader has to refer to the table to see if they were significant. The reporting is not very consistent in this respect. Can these results be added alongside the others? 10. “The chance of induction of labour” should be “the odds of induction of labour” 11. Final line on page 11 “and not obese (OR 0.96.....)” should be “and obese (OR 0.96.....)” i.e. remove not 12. Table 3 It would be informative to see a column showing the amount of missing data present for each outcome variable rather than the reader trying to work it out from the table. Can these be added? 13. The authors state that each outcome was additionally adjusted for conditions that precede or can occur alongside it. I think it would be useful to reflect this in the footnotes of table 3 i.e. clarify in the footnotes which variables were used in each of the different analyses
--	--

	14. Finally I think that footnotes 2 and 3 in table 3 are pretty important and I think it would be good if the authors could include a brief sentence in the main body of the text to highlight that these cases were lost. Additionally, is there a clinical reason why so many of the post term births did not have these outcomes? Could this have caused any bias? 15. Discussion My other major outstanding comment regards the discussion. I think the authors need to revisit the discussion again because not all of the outcomes appear to be adequately discussed. For example the authors include a relatively large amount of discussion around caesarean section however there is no discussion of other variables e.g. induction of labour, pre/post term delivery etc. I would like to see a more balanced discussion including all of the outcomes that the authors investigated with reference back to previous literature. Can the authors ensure that they have covered each of the outcomes in the discussion. 16. Strength and limitations The authors state “adequate adjustment for confounders”. Again this is not accurate given the study is missing several potentially important confounders. The authors have highlighted this as a limitation and later state that they have “a limited set of confounders”. Please therefore remove adequate from the main box and replace with something more appropriate. 17. The authors state that they combined underweight with normal weight but that the % of underweight was small. Could you please state what the percentage of underweight was? 18. “Using this as reference group” should be “using this as the reference group” 19. The authors state that “classifying morbid obesity would have given greater additional insight..”. Could you add that this wasn’t done because the data were not available (rather than you just didn’t do it) 20. P15 typo – last two sentences of strength and limitations section “time of birth were not available in dataset” – add the word ‘the’ before dataset 21. “Could not use neonatal outcomes“ could be better phrased “could not analyse neonatal outcomes” 22. Funding Farr institute @ Scotland doesn’t exist anymore. Should this now be HDR UK?? 23. The attached STROBE checklist
--	---

	This is fine however I note that several of the cited page numbers are inaccurate and need to be updated to reflect the latest draft
--	--

VERSION 3 – AUTHOR RESPONSE

Response to reviews

Reviewer 2	
The authors have redone the analyses using the full cohort rather than the staged design and this is appropriate. I think it is a pity that some of the results for certain outcomes - stillbirth, postpartum hemorrhage and NICU admission have been omitted for the reason of the outcome assessment being incomplete, as they show similar results as other studies (if anything, an incomplete outcome assessment would most likely have led to underestimation of the observed associations). Other than that I don't have any further comments.	Thanks for your comments and we are happy that this reviewer is ok with the analysis. Using the full cohort rather than the staged approach means some outcomes had to be dropped due to inadequate case ascertainment as suggested by reviewer 5.
Reviewer 5	
1. Introduction My major outstanding comment is that the introduction no longer adequately reflects the paper. The authors focus their introduction on stillbirths, neonatal/perinatal/infant deaths and admissions to neonatal units; however, the authors have removed these outcomes from the manuscript and so they are no longer relevant. Similarly, with the exception of one sentence mentioning diabetes, and hypertension, the introduction does not actually mention any of the other outcomes that are studied. The introduction states that no recent study in Scotland has investigated the effect of high BMI on pregnancy and delivery outcomes. Does this mean that there are some studies in Scotland but none more recently? Or none at all? What does more recently mean? What about studies worldwide? In light of the fact many of the original outcomes have been removed from the paper, I think the authors need to revisit the introduction in order to make it more appropriate and relevant to the rest of the paper. In particular, I think the reader needs to see a little bit more about what previous studies worldwide have found with respect to associations with each of the outcomes.	The introduction has been significantly revised to address the reviewer's concern.
2. Strengths and limitations box The authors state that they have used a "national database covering all major maternal and neonatal	We have addressed this and it now reads, "covering some of the major maternal and

outcomes in Scotland over eight years". In light of several of the most major outcomes (stillbirths, congenital anomalies etc.) having actually been removed this is no longer true. Please change the wording to something more appropriate.	neonatal outcomes in Scotland over eight recent years."
3. In the next sentence, the authors state "adequate adjustment for confounders". Again this is not accurate given the study is missing several potentially important confounders. The authors have highlighted this as a limitation and later state that they have "a limited set of confounders". Please therefore remove adequate from the main box and replace with something more appropriate.	We have addressed this and it now reads: "Analysis used whole study population with adjustment for some confounders to estimate impact of high maternal-weight status on each outcome".
4. Methods Minor comment/observation - I do not really understand the term "clinical audit" in this context. Surely, this is just a regular retrospective cohort study using routinely collected administrative datasets?	John/Louise/Andrew – should we remove the term clinical audit? If so, what is the implication of this on the ethics approval statement in the methods section?
5. Minor comment/suggestion - Table 1 is very small and so perhaps the definitions of BMI could just be described as text within the body of the paper rather than a separate table?	We have now removed Table 1 and described this as text within the body of the paper.
6. It would be informative for the reader to see an additional line in table 2 showing the number and percentage of missing data for each of the covariates. Smoking in particular is renowned as having rather a lot of missing data. How did the authors deal with missing data? Were women with missing data just dropped from the analyses or did you think about imputation?	We have included a supplementary file showing missing data for both covariates and all outcomes.
7. Data analysis The authors state that each outcome was additionally adjusted for conditions that precede or can occur alongside it. I think it would be useful to reflect this within the footnotes of table 3 i.e. can you clarify in the footnotes which variables were used in each of the different analyses?	We realized that this was a bit complicated to do as some conditions are mutually exclusive. Therefore, we have revised the footnote and amended the table which shows all the required information.
8. Results In the second paragraph, the sentence beginning "the risk of the three conditions...." should be made clearer for the reader. Could this be rephrased as "the risk of gestational diabetes, preeclampsia and hypertension....."	We have amended the sentence. It now reads: "The risk of gestational diabetes, pre-eclampsia and hypertension increased steadily with increasing BMI."

9. Furthermore, the authors have not discussed the results for hypertension, pre or post term delivery or induction of labour in the results text meaning that the reader has to refer to the table to see if they were significant. The reporting is not very consistent in this respect. Can these results be added alongside the others?	We have now discussed the results of all remaining outcomes highlighted by the reviewer.
10. "The chance of induction of labour" should be "the odds of induction of labour"	We have addressed this.
11. Final line on page 11 "and not obese (OR 0.96.....)" should be "and obese (OR 0.96.....)" i.e. remove not	We have removed 'not' from the sentence.
12. Table 3 It would be informative to see a column showing the amount of missing data present for each outcome variable rather than the reader trying to work it out from the table. Can these be added?	We have included a supplementary file showing missing data for both covariates and all outcomes.
13. The authors state that each outcome was additionally adjusted for conditions that precede or can occur alongside it. I think it would be useful to reflect this in the footnotes of table 3 i.e. clarify in the footnotes which variables were used in each of the different analyses	Andrew/Louise?
14. Finally I think that footnotes 2 and 3 in table 3 are pretty important and I think it would be good if the authors could include a brief sentence in the main body of the text to highlight that these cases were lost. Additionally, is there a clinical reason why so many of the post terms births did not have these outcomes? Could this have caused any bias?	Andrew/Louise?
15. Discussion My other major outstanding comment regards the discussion. I think the authors need to revisit the discussion again because not all of the outcomes appear to be adequately discussed. For example the authors	The discussion has been revised to address to cover all each of the outcomes examined in the paper.

include a relatively large amount of discussion around caesarean section however there is no discussion of other variables e.g. induction of labour, pre/post term delivery etc. I would like to see a more balanced discussion including all of the outcomes that the authors investigated with reference back to previous literature. Can the authors ensure that they have covered each of the outcomes in the discussion.	
16. Strength and limitations The authors state “adequate adjustment for confounders”. Again this is not accurate given the study is missing several potentially important confounders. The authors have highlighted this as a limitation and later state that they have “a limited set of confounders”. Please therefore remove adequate from the main box and replace with something more appropriate.	We have replaced ‘adequate’ with ‘limited’.
17. The authors state that they combined underweight with normal weight but that the % of underweight was small. Could you please state what the percentage of underweight was?	We were informed that the percentage of underweight was negligible. Unfortunately, we didn’t receive the actual percentage information from the meta data so will not be able to provide this.
18. “Using this as reference group” should be “using this as the reference group“	We have addressed this.
19. The authors state that “classifying morbid obesity would have given greater additional insight..”. Could you add that this wasn’t done because the data were not available (rather than you just didn’t do it)	We have explained that the dataset we received did not differentiate the categories of obesity and it was not possible to do this retrospectively, so all women with BMI of 30 or more were considered as having obesity.
20. P15 typo – last two sentences of strength and limitations section “time of birth were not available in dataset” – add the word ‘the’ before dataset	We have addressed this.
21. “Could not use neonatal outcomes“ could be better phrased “could not analyse neonatal outcomes”	We have addressed this.
22. Funding Farr institute @ Scotland doesn’t exist anymore. Should	Farr Institute is still the correct term.

this now be HDR UK??	
23. The attached STROBE checklist This is fine however I note that several of the cited page numbers are inaccurate and need to be updated to reflect the latest draft	Larry - to do this once ready to submit

VERSION 4 – REVIEW

REVIEWER	Michael Fleming University of Glasgow Scotland United Kingdom
REVIEW RETURNED	05-Dec-2019

GENERAL COMMENTS	This manuscript is much improved and I am happy that the authors have actioned the majority of my previous review comments and concerns. I still have a number of comments however which I feel need to be addressed before the manuscript will be suitable for publication. These are outlined below in no particular order:  - Table 3 - My main concern is around table 3 because I am still unclear as to why the different sample sizes arose in each of the respective analyses and which variables were adjusted for each time. I am unclear as to how table 3 ties in with the supplementary file 2 which I assume is the flow diagram? I find that I need to constantly go between the two in order to try and figure out what has been adjusted for and why records have been lost and in the majority of cases I am unsuccessful. I think this needs to be presented much more clearly and concisely. I still think the table could also benefit from additional footnotes and that some additional text in the manuscript would be beneficial to the reader in order to fully clarify this. I also still cannot work out why the sample for the post term analysis is so low. - With further reference to supplementary file 2 can the authors briefly explain in the manuscript why they chose to only look at first time singleton mothers and also why they chose to look at mothers between 20 and 40 years of age. In particular why were teenage pregnancies excluded? - The introduction is now much improved and references many of the outcomes that are investigated in the paper. However I still see no reference to some of the outcomes e.g. Apgar score, induction of labour, post terms delivery, placenta praevia, placental abruption. What are the authors reasons for looking at those? A little bit of background on previous literature/findings for these outcomes is also needed to help orient the reader and allow them to understand why these outcomes are important and why they merit investigation
---

	 - I would like to see a brief statement of the authors initial research hypotheses - what did the researchers expect to see regarding these outcomes? - I may have missed this but I did not see a statement regarding ethics - I am not keen on the term 'whole study population' and I think using a term such as 'population-wide study' or something similar would be much clearer for the reader to understand - What is meant by the term 'essential hypertension' and how does this differ from hypertension? This is not clear - Page 6 - The authors wrongly state that the outcome variables are recorded according to ICD10 - only some of the outcomes (e.g. preeclampsia/hypertension) would have been recorded in this way but many (e.g. apgar score/estimated gestation) wouldn't. Can this statement be corrected. - Throughout the manuscript the authors use 'analysis' when the plural 'analyses' should be used instead since many separate analyses were carried out - Page 6 - authors state that 80% of pregnant mothers present for antenatal care - do you have a reference for this? - Table 1 - Could the authors have not just have shown demographics for the 129,773 women who had complete data on all of the predictors and delivery outcomes? Rather than showing the larger number with complete predictor data only and then having to state that the number included in the analyses is actually smaller after removing those with no outcome data? This is a bit confusing and actually the results regardless will probably be the same - Results section - The authors jump between using terms such as 'risk', 'odds', 'odds ratio', 'OR', 'ORs', and even 'risk ratio' when describing the effect size. Terminology needs to be accurate and consistent. These are odds ratios therefore the term odds should be used rather than risk and certainly not risk ratio which is incorrect. I suggest defining the terms odd ratio (OR) and then sticking to odds throughout or odds ratio if appropriate - page 12 - The authors have presented two sets of odds ratios and CIs for LGA outcome pertaining to overweight women (1.27 and 1.30) - the second one should be deleted - The authors have described the effect sizes for the pre term and post term outcomes however actually only the pre term outcome for the obese group provided a significant result - the others were not significant. This has been mentioned in the discussion section but it should also be mentioned in the results section for clarity. - The authors could maybe think about showing p values throughout particularly for outcomes such as apgar score which was barely statistically significant
--	--

	- strengths/limitations section - the authors state that only a small percentage of women were underweight in Scotland in recent years - are there any references for this?- There are various typographical errors throughout the manuscript which need to be addressed. Additionally please watch use of past and present tense
--	--

VERSION 4 – AUTHOR RESPONSE

Reviewer 5	
Table 3 - My main concern is around table 3 because I am still unclear as to why the different sample sizes arose in each of the respective analyses and which variables were adjusted for each time. I am unclear as to how table 3 ties in with the supplementary file 2 which I assume is the flow diagram? I find that I need to constantly go between the two in order to try and figure out what has been adjusted for and why records have been lost and in the majority of cases I am unsuccessful. I think this needs to be presented much more clearly and concisely. I still think the table could also benefit from additional footnotes and that some additional text in the manuscript would be beneficial to the reader in order to fully clarify this. I also still cannot work out why the sample for the post term analysis is so low.	We have brought the table from the supplemental files into the manuscript as this clearly sets out the list of covariates in each model. The analysis section of the manuscript has been re-written in an attempt to make the situation clearer. As previously explained the post-term delivery sample is small and ends up being dropped from many of the models as pregnancies experiencing complications such as placenta praevia or a large for gestational age baby are unlikely within the NHS to progress beyond term without being induced or delivered another way. The other differences in sample sizes are due to the fact that some of the outcomes are mutually exclusive (induction, c-section, emergency c-section) and therefore it would be incoherent to adjust a model for a mutually exclusive outcome (e.g. adjusting small for gestational age for large for gestational age). We hope these changes now make the situation clearer. While Table 3 is complex the reporting requirements of STROBE require the inclusion of all the numbers.
With further reference to supplementary file 2 can the authors briefly explain in the manuscript why they chose to only look at first time singleton mothers and also why they chose to look at mothers between 20 and 40 years of age. In particular why were teenage pregnancies excluded?	In order to avoid women being included more than once if they had more than one birth, we restricted the analysis to first time singleton mothers to ensure that the births in the sample are relatively independent. This was stated under strengths and limitations section of the manuscript. We originally had the following age categories in our first submission on 22 August 2018 – “15-19”, “20-24”, “25-29”, “30-34”, “35-39”, “40-44”, “45-49”. However, this reviewer rightly noted that for several of the outcomes, no cases or very few cases were encountered in some age groups. This meant that for several of the outcomes reduced obesity cell sizes made the analysis less than optimally powerful, thus they lacked power for the study outcomes. In order to address this we limited the age range to 20-40 to address the issue of small numbers of cases within some age groups. We have added a sentence to this effect in the limitations.
The introduction is now much improved and references many of the outcomes that are investigated in the paper. However I still see no reference to some of the outcomes e.g. Apgar score, induction of labour, post terms delivery, placenta praevia, placental	We have mentioned these outcomes in the introduction and provided a rationale for looking at them. They have also been discussed further in the discussion section of the manuscript.

abruption. What are the authors reasons for looking at those? A little bit of background on previous literature/findings for these outcomes is also needed to help orient the reader and allow them to understand why these outcomes are important and why they merit investigation	
I would like to see a brief statement of the authors initial research hypotheses - what did the researchers expect to see regarding these outcomes?	We have provided our initial hypothesis in the introduction.
I may have missed this but I did not see a statement regarding ethics	The study was designed as a clinical audit so did not require approval from a Research Ethics Committee. However, approval was obtained from the Public Benefit and Privacy Panel via the national Electronic Data Research and Innovation Service to use the anonymised data collected by these registries. We have added this statement to the manuscript.
I am not keen on the term 'whole study population' and I think using a term such as 'population-wide study' or something similar would be much clearer for the reader to understand	We have replaced 'whole study population' to 'population-wide study'.
What is meant by the term 'essential hypertension' and how does this differ from hypertension? This is not clear	ICD-11 code BA00 for Essential hypertension gives the following definition: 'Essential (primary) hypertension, accounting for 95% of all cases of hypertension, is defined as high blood pressure for which a secondary cause cannot be found.' As the authors of the study being cited use the term essential hypertension we do not feel that it is appropriate to alter the term.
Page 6 - The authors wrongly state that the outcome variables are recorded according to ICD10 - only some of the outcomes (e.g. preeclampsia/hypertension) would have been recorded in this way but many (e.g. appgar score/estimated gestation) wouldn't. Can this statement be corrected.	We have amended the statement to indicate that 'relevant' outcome variables are recorded according to ICD10. We have also added a reference to the definitions for the terms not defined by ICD.
Throughout the manuscript the authors use 'analysis' when the plural 'analyses' should be used instead since many separate	We have replaced relevant 'analysis' term to 'analyses'

analyses were carried out	
Page 6 - authors state that 80% of pregnant mothers present for antenatal care - do you have a reference for this?	We have added a reference to the statement.
Table 1 - Could the authors have not just have shown demographics for the 129,773 women who had complete data on all of the predictors and delivery outcomes? Rather than showing the larger number with complete predictor data only and then having to state that the number included in the analyses is actually smaller after removing those with no outcome data? This is a bit confusing and actually the results regardless will probably be the same	Thank you for your suggestion, we have made this change.
Results section - The authors jump between using terms such as 'risk', 'odds', 'odds ratio', 'OR', 'ORs', and even 'risk ratio' when describing the effect size. Terminology needs to be accurate and consistent. These are odds ratios therefore the term odds should be used rather than risk and certainly not risk ratio which is incorrect. I suggest defining the terms odd ratio (OR) and then sticking to odds throughout or odds ratio if appropriate	We have replaced 'risk' with odds or odds ratio where appropriate.
Page 12 - The authors have presented two sets of odds ratios and CIs for LGA outcome pertaining to overweight women (1.27 and 1.30) - the second one should be deleted	Sorry for the oversight – we have now removed the second OR and CI.
The authors have described the effect sizes for the pre term and post term outcomes however actually only the pre term outcome for the obese group provided a significant result - the others were not significant. This has been mentioned in the discussion section but it should also be mentioned in the results section for clarity.	We have included in the results section that regarding the odd ratios of pre-term and post-term outcomes only the pre-term outcome for the obese group was significant and the others were not significant.

The authors could maybe think about showing p values throughout particularly for outcomes such as appgar score which was barely statistically significant	AJW: P-values provide less information than the confidence intervals provided and are being heavily criticised at the moment. Equating clinical significance with a somewhat arbitrary statistical cut-point is especially challenging in population wide research where small effects for individuals can have large population effects. We don't feel that it is appropriate to include p-values in this instance.
Strengths/limitations section - the authors state that only a small percentage of women were underweight in Scotland in recent years - are there any references for this?	We have added in a reference to support this statement.
There are various typographical errors throughout the manuscript which need to be addressed. Additionally please watch use of past and present tense	We have read through the manuscript and addressed all the errors.

VERSION 5 – REVIEW

REVIEWER	Michael Fleming University of Glasgow UK
REVIEW RETURNED	23-Jan-2020

GENERAL COMMENTS	I think this manuscript reads very well and will be of great interest on publication. I am satisfied that the authors have suitably addressed all of my comments and I am happy to accept this manuscript as being suitable for publication. I would like to thank the authors for addressing my comments throughout the review process. I think the current version reads well and the authors have done a great job with improving the clarity of the paper throughout over the last few iterations. I now feel that the introduction suitably covers each of the outcomes and the authors' reasons for analyzing them before clearly stating the research hypotheses. The discussion then covers all of the outcomes, suitably contextualizing the findings using previous literature. The analyses undertaken and the results reported are now a lot clearer as are the various tables which complement each other nicely and aid understanding. I have no further concerns however I do just want to flag up some very minor typos which I noticed. Page 4 - "2003 to 2010 examined the impact" should be "2003 to 2010 examining the impact" Page 6 - and in the supplemental file 1.. - delete the extra . Page 14 - second line - "statistically significant different" should be "statistically significantly different" Page 18 - second last line - "The analyses used a population wide data" should be "The analyses used population wide data" Finally I would like to apologize to the authors for my comment querying why they decided to only look at women aged between 20-40 thus omitting teenage pregnancies. The authors correctly reminded me that it was actually me who suggested (in a previous
---

	review) that they be removed due to small numbers occurring in these groups! So I apologize for the confusion around that point and thank the authors for reminding me of the reasons for this!
--	---